# Evaluating Predictive Distributions: Does Bayesian Deep Learning Work?

## Abstract

Posterior predictive distributions quantify uncertainties ignored by point estimates. This paper introduces *The Neural Testbed*, which provides tools for the systematic evaluation of agents that generate such predictions. Crucially, these tools assess not only the quality of marginal predictions per input, but also joint predictions given many inputs. Joint distributions are often critical for useful uncertainty quantification, but they have been largely overlooked by the Bayesian deep learning community. We benchmark several approaches to uncertainty estimation using a neural-network-based data generating process. Our results reveal the importance of evaluation beyond marginal predictions. Further, they reconcile sources of confusion in the field, such as why Bayesian deep learning approaches that generate accurate marginal predictions perform poorly in sequential decision tasks, how incorporating priors can be helpful, and what roles epistemic versus aleatoric uncertainty play when evaluating performance. We also present experiments on real-world challenge datasets, which show a high correlation with testbed results, and that the importance of evaluating joint predictive distributions carries over to real data. As part of this effort, we opensource The Neural Testbed, including all implementations from this paper.

## 1 Introduction

Deep learning has emerged as the state-of-the-art approach across a number of application domains in which agents learn from large amounts of data (LeCun et al., 2015). Neural networks are increasingly used not only to predict outcomes but also to inform decisions. Common approaches in deep learning produce point estimates but not uncertainty estimates, which are often required for effective decision-making. Bayesian deep learning extends the methodology to produce such uncertainty estimates (MacKay, 1992; Neal, 2012).

We consider agents that are trained on data pairs $((X_t, Y_{t+1}) : t = 0, 1, \ldots, T-1)$ and subsequently generate predictions given new inputs. When presented with an input $X_T$, a Bayesian neural network can generate a predictive distribution of the outcome $Y_{T+1}$ that is yet to be observed. This distribution characterizes the agent's uncertainty about $Y_{T+1}$. We refer to such a prediction as *marginal* to distinguish it from a joint predictive distribution over a list $(Y_{T+1}, \ldots, Y_{T+\tau})$ of prospective outcomes corresponding to inputs $(X_T, \ldots, X_{T+\tau-1})$.

The importance of uncertainty estimation has motivated a great deal of research over recent years (Kendall & Gal, 2017). This research has produced a variety of agents that learn to generate predictive distributions. With this proliferation of alternatives, it is increasingly important to analyze and compare their performance (Filos et al., 2019; Nado et al., 2021). In this paper, we introduce new tools for systematic evaluation of such agents.

Our tools overcome several limitations faced by previous methods of evaluation. First, by focusing purely on *predictive* distributions, we allow for a unified treatment of approaches developed within the Bayesian neural network community and beyond. This sidesteps the

---

Open source code available at https://anonymous.4open.science/r/neural-testbed-B839.

question of whether any approach 'is *really* Bayesian' (Wilson & Izmailov, 2020). Second, our tools evaluate the quality of higher-order joint predictions ($\tau > 1$). Until now, the Bayesian deep learning literature has focused almost exclusively on evaluating marginal predictions (Wang et al., 2021). Finally, we develop a neural-network-based data generating process for Bayesian deep learning that can be used to drive insight and algorithm development. Where research has focused on a small set of challenge datasets, this might introduce bias through overfitting via multiple iterations of algorithm development. We use these tools to compare hundreds of agent variants. Further, we show that performance on our synthetic data generating process data is highly correlated with performance on real-world challenge datasets. We opensource all code used in this paper as *The Neural Testbed*.

Our results reconcile several sources of confusion in the field. One concerns why particular approaches developed by the Bayesian deep learning community, such as Bayes-by-backprop, dropout, and deep ensembles, perform poorly in sequential decision tasks despite faring well based on evaluation metrics of that community (Osband et al., 2018). Our results demonstrate that, while such methods produce accurate marginal predictions, they are no longer competitive when it comes to high-order joint predictions. Joint predictions play a critical role in sequential decision-making (Lu et al., 2021).

Another puzzling issue is that state-of-the-art methods do not employ domain-specific priors. Whether Bayesian deep learning approaches should at all is a subject of controversy (Wenzel et al., 2020). We show that the benefits of domain-specific priors can be pronounced when evaluating high-order joint predictions, even where they are negligible for marginals.

We also help to resolve a point of philosophical debate within the deep learning community: the importance of epistemic versus aleatoric uncertainty[1]. The strangeness of this distinction has even made its way into wider popular culture, as satirized in the XKCD comic of Figure 1 (Munroe, 2021). For a given parametric model, we can clearly distinguish parameter uncertainty from noise, or reducible from irreducible uncertainty. However, from the perspective of a learning agent, the choice of model is subjective; different models can lead to the same marginal predictions. Our formulation provides a clear and objective way to assess the quality of predictive distributions, without reliance on this subjective distinction between knowledge and chance. Crucially, we show that this can be judged via the quality of joint predictions, but that marginals are not sufficient.

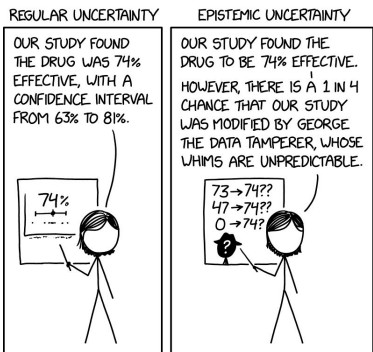

Figure 1: Epistemic or aleatoric? Does it matter?

It is worth mentioning another notable contribution of this work. The quality of a predictive distribution is commonly assessed in terms of cross-entropy loss. While this measure is well-defined for both marginal and joint predictions, to the best of our knowledge, the literature has only addressed computation in the former case. For high-order joint predictions, the straightforward approach would require computing sums over exponentially many values. To render this computationally tractable, we developed a novel approximation algorithm that leverages a random partitioning operation and Monte Carlo simulation. While this approach is motivated by concepts from high-dimensional geometry (Kaski, 1998; Donoho, 2006), we leave its analysis as a topic for future theoretical research.

---

[1] *Epistemic* uncertainty relates to knowledge (ancient Greek *episteme*↔knowledge), as opposed to *aleatoric* uncertainty relating to chance (Latin *alea*↔dice) (Der Kiureghian & Ditlevsen, 2009).

## 2 Evaluating predictive distributions

In this section, we introduce notation for the standard supervised learning framework we will consider (classification) as well as our evaluation metric (the KL-loss). We also explain how we estimate the KL-loss for high-order joint predictions where exact computation is infeasible, using random partitions and Monte Carlo simulation.

### 2.1 Kullback–Leibler loss

Consider a sequence of pairs $((X_t, Y_{t+1}) : t = 0, 1, 2, \ldots)$, where each $X_t$ is a feature vector and each $Y_{t+1}$ is its target label. This sequence is i.i.d. conditioned on the environment $\mathcal{E}$, which produces the data, and which we view as a latent random variable. We consider an agent that is uncertain about the environment and predicts class labels $Y_{T+1:T+\tau} \equiv (Y_{T+1}, \ldots, Y_{T+\tau})$ given training data pairs $\mathcal{D}_T \equiv ((X_t, Y_{t+1}) : t = 0, 1, 2, \ldots, T - 1)$ and unlabelled feature vectors $X_{T:T+\tau-1} \equiv (X_T, \ldots, X_{T+\tau-1})$. From the agent's perspective, each feature vector $X_t$ is generated i.i.d from a fixed distribution $\mathbb{P}(X_t \in \cdot)$, and each class label $Y_{t+1}$ is then drawn from $\mathbb{P}(Y_{t+1} \in \cdot | \mathcal{E}, X_t)$.

We describe the agent's predictions in terms of a generative model, parameterized by a vector $\theta_T$ that the agent learns from the training data $\mathcal{D}_T$. For any inputs $X_{T:T+\tau-1}$, $\theta_T$ determines a predictive distribution, which could be used to sample imagined outcomes $\hat{Y}_{T+1:T+\tau}$. We define the $\tau^{\text{th}}$-order expected KL-loss by

$$\mathbf{d}_{\text{KL}}^{\tau} = \mathbb{E}\Big[\mathbf{d}_{\text{KL}}\Big(\underbrace{\mathbb{P}\left(Y_{T+1:T+\tau} \in \cdot | \mathcal{E}, X_{T:T+\tau-1}\right)}_{\text{environment likelihood}} \Big\| \underbrace{\mathbb{P}(\hat{Y}_{T+1:T+\tau} \in \cdot | \theta_T, X_{T:T+\tau-1})}_{\text{agent likelihood}}\Big)\Big] \quad (1)$$

$$= \underbrace{-\mathbb{E}\Big[\log\Big(\mathbb{P}\Big(\hat{Y}_{T+1:T+\tau} = Y_{T+1:T+\tau}\Big|\theta_T, X_{T:T+\tau-1}, Y_{T+1:T+\tau}\Big)\Big)\Big]}_{\text{cross-entropy loss} \equiv \text{negative log-likelihood}} + C,$$

where $C = \mathbb{E}\left[\log\left(\mathbb{P}\left(Y_{T+1:T+\tau} | \mathcal{E}, X_{T:T+\tau-1}\right)\right)\right]$ is independent of $\theta_T$. The expectation is taken over all random variables, including the environment $\mathcal{E}$, the parameters $\theta_T$, $X_{T:T+\tau-1}$, and $Y_{T+1:T+\tau}$. Note that $\mathbf{d}_{\text{KL}}^{\tau}$ is equivalent to the widely used notion of cross-entropy loss, though offset by a quantity that is independent of $\theta_T$ (Kullback & Leibler, 1951). For $\tau > 1$, $\mathbf{d}_{\text{KL}}^{\tau}$ assesses joint rather than the marginal predictions.

### 2.2 Marginal Versus Joint Predictions

Evaluating an agent's ability to estimate uncertainty on joint instead of marginal predictions can result in very different answers. We provide a simple example that illustrates the point. Suppose the data $((X_t, Y_{t+1}) : t = 0, 1, 2, \ldots)$ is generated by repeated tosses of a possibly biased coin with unknown probability $p$ of heads.[2] Let $X_t = 0$, to indicate that there is no input, and let each outcome $Y_{t+1}$ be 0 or 1 to indicate tails or heads, respectively. Consider two agents that, without any training, predict outcomes. Agent 1 assumes $p = 2/3$ and models the outcome of each flip as pure chance. Agent 2 assumes that the coin is fully biased, meaning that $p \in \{0, 1\}$, but assigns probabilities 1/3 and 2/3 to 0 and 1.

Let $\hat{Y}_{t+1}^1$ and $\hat{Y}_{t+1}^2$ denote the outcomes imagined by the two agents. Despite their differing assumptions, the two agents generate identical marginal predictive distributions: $\mathbb{P}(\hat{Y}_{t+1}^1 = 0) = \mathbb{P}(\hat{Y}_{t+1}^2 = 0) = 1/3$. On the other hand, joint predictions greatly differ for large $\tau$:

$$\mathbb{P}(\hat{Y}_1^1 = 0, .., \hat{Y}_\tau^1 = 0) = 1/3^\tau \ll 1/3 = \mathbb{P}(\hat{Y}_1^2 = 0, \ldots, \hat{Y}_\tau^2 = 0).$$

We can say that agent 1 attributes all uncertainty to aleatoric sources and agent 2, epistemic sources (although as Figure 1 alludes, there are many ways an agent can attribute sources of uncertainty). Evaluating marginal predictions cannot distinguish between the two possibilities, though for a specific prior distribution over $p$, one agent could be right and the other wrong. One must evaluate joint predictions to make this distinction.

---

[2]We consider this coin as an illustrative model of more complex binary outcomes, such as whether a user will click on an ad, or whether a given mortgage will default on payments.

When it comes to decision-making, this distinction can be critical (Lu et al., 2021). In a casino, under the first agent's assumption, there is large upside and little risk on repeatedly betting on heads in the long run. However, if there is a $1/3$ chance the coin will *always* land tails, as is the case in the second agent's prediction, there is a ruinous risk to repeatedly betting heads. Evaluating joint predictions beyond marginals distinguishes these cases.

### 2.3 Computation of Kullback–Leibler loss

In contexts we will consider, it is not possible to compute $\mathbf{d}_{\mathrm{KL}}^{\tau}$ exactly. As such, we will approximate $\mathbf{d}_{\mathrm{KL}}^{\tau}$ via Monte Carlo simulation. This section provides a high level overview of our approach, we push the full details to Appendix A. Algorithm 1 outlines a basic approach to estimating $\mathbf{d}_{\mathrm{KL}}^{\tau}$ with respect to a synthetic data generating process. The algorithm samples a set of environments and a training dataset for each environment. For each of these pairs, the agent is re-initialized, trained, and then tested on $N$ independent test data $\tau$-samples. Note that each test data $\tau$-sample includes $\tau$ data pairs. For each test data $\tau$-sample, the likelihood of the environment is computed exactly, but that of the agent's belief distribution is approximated. The estimate of $\mathbf{d}_{\mathrm{KL}}^{\tau}$ is taken to be the sample mean of the log-likelihood-ratios (Algorithm 2).

---

**Algorithm 1** KL-Loss Computation

---

1: **for** $j = 1, 2, \ldots, J$ **do**
2:      sample environment and training dataset, and train agent
3:      **for** $n = 1, 2, \ldots, N$ **do**
4:          sample a test data $\tau$-sample with $\tau$ feature-label pairs
5:          compute $p_{j,n}$                               ▷ likelihood of environment
6:          compute $\hat{p}_{j,n}$               ▷ estimated likelihood of agent's belief distribution
7: **return** $\frac{1}{JN} \sum_{j=1}^{J} \sum_{n=1}^{N} \log\left(p_{j,n} / \hat{p}_{j,n}\right)$          ▷ estimated log-likelihood-ratio

---

While the likelihood of an environment can be efficiently computed, that of an agent's belief distribution poses a computational challenge. One approach is to estimate this likelihood via Monte Carlo simulation (Algorithm 3). This produces unbiased estimates, which can be accurate when $\tau$ is small. However, maintaining accuracy requires the number of samples to grow exponentially with $\tau$, as discussed in Appendix A.1. To overcome this challenge, we propose a novel approach that estimates the likelihood of the agent's beliefs via a combination of randomized partitioning and Monte Carlo simulation (Algorithm 4) (Kaski, 1998). We conjecture that, under suitable regularity conditions, this novel approach produces accurate estimates even when $\tau$ is large, but leave a formal analysis to future work. Even though Algorithm 1 is developed for a synthetic data generating process, it is straightforward to extend it to evaluate agents on real data. We outline our approach to real data in Section 5.1, with full details in Appendix A.2.

## 3 Benchmark agents

In this section we outline the baseline agents that we use to benchmark canonical approaches to uncertainty estimation in deep learning. Table 1 links to papers that introduce these agents, as well as the hyperparamters that we tuned to optimize their performance via gridsearch. In each case, we attempt to match 'canonical' implementations, which we open source at https://anonymous.4open.science/r/neural-testbed-B839.

In addition to these agent implementations, our opensource project contains all the evaluation code to reproduce the results of this paper. Our code is written in Python and makes use of Jax internally (Bradbury et al., 2018). However, our evaluation procedure is framework agnostic, and can equally be used with any Python package including Tensorflow, Pytorch or even SKlearn. Over the course of this paper, we have made extensive use of parallel computation to facilitate large hyperparameter sweeps over many problems. Nevertheless, the overall computational cost is relatively low by modern deep learning standards and relies only on standard CPU. For reference, evaluating the `mlp` agent across all the problems in

| agent | description | hyperparameters |
|-------|-------------|-----------------|
| mlp | Vanilla MLP | $L_2$ decay |
| ensemble | 'Deep Ensemble' (Lakshminarayanan et al., 2017) | $L_2$ decay, ensemble size |
| dropout | Dropout (Gal & Ghahramani, 2016) | $L_2$ decay, network, dropout rate |
| bbb | Bayes by Backprop (Blundell et al., 2015) | prior mixture, network, early stopping |
| sgmcmc | Stochastic Langevin MCMC (Welling & Teh, 2011) | learning rate, prior, momentum |
| ensemble+ | Ensemble + prior functions (Osband et al., 2018) | $L_2$ decay, ensemble size, prior scale, bootstrap |
| hypermodel | Hypermodel (Dwaracherla et al., 2020) | $L_2$ decay, prior, bootstrap, index dimension |

Table 1: Summary of benchmark agents, full details in Appendix B.

our testbed and real data requires less than 3 CPU-hours. We view our opensource effort as one of the major contributions of this work. We provide clear and strong baselines, together with an objective and accessible method for assessing uncertainty estimates beyond marginal distributions.

## 4    THE NEURAL TESTBED

In this section we introduce the Neural Testbed, a system for assessing and comparing agent performance. The Testbed implements synthetic data generating processes and streamlines the process of sampling data, training agents, and evaluating test performance by estimating KL-loss for marginal and high-order joint predictions. Since independent data can be generated for each execution, the Testbed can drive insight and multiple iterations of algorithm development without risk of overfitting to a fixed dataset. We begin by describing the simple generative model based around a random 2-layer MLP. We then apply this testbed to evaluate a comprehensive set of benchmark agents.

### 4.1    SYNTHETIC DATA GENERATING PROCESSES

By data generating process, we do not mean only the conditional distribution of data pairs $(X_t, Y_{t+1})|\mathcal{E}$ but also the distribution of the environment $\mathcal{E}$. The Testbed considers 2-dimensional inputs and binary classification problems, although the generating processes can be easily extended to any input dimension and number of classes. The Testbed offers three data generating processes distinguished by a "temperature" setting, which signifies the signal-to-noise ratio (SNR) regime of the generated data. The agent can be tuned separately for each setting. This reflects prior knowledge of whether the agent is operating in a high SNR regime such as image recognition or a low SNR regime such as weather forecasting.

To generate a model, the Testbed samples a 2-hidden-layer ReLU MLP with 2 output units, which are scaled by 1/temperature and passed through a softmax function to produce class probabilities. The MLP is sampled according to standard Xavier initialization (Glorot & Bengio, 2010), with the exception that biases in the first layer are drawn from $N(0, \frac{1}{2})$. The inputs $(X_t : t = 0, 1, \ldots)$ are drawn i.i.d. from $N(0, I)$. The agent is provided with the data generating process as prior knowledge.

In Section 2.1, we described KL-loss as a metric for evaluating performance of an agent. The Neural Testbed estimates KL-loss, with $\tau \in \{1, 100\}$, for three temperature settings and several training dataset sizes. For each value of $\tau$, the KL-losses are averaged to produce an aggregate performance measure. Further details concerning data generation and agent evaluation are offered in Appendix A.

### 4.2    PERFORMANCE IN MARGINAL PREDICTIONS

We begin our evaluation of benchmark approaches to Bayesian deep learning in marginal predictions ($\tau = 1$). This setting has been the main focus of the Bayesian deep learning literature. Despite this focus, it is surprising to see in Figure 2 that none of the benchmark methods significantly outperform a well-tuned MLP baseline according to $\mathbf{d}_{\mathrm{KL}}^1$. Of course, there are many other metrics one might consider, but in this fundamental metric of prediction quality, the mlp agent presents a baseline that is difficult to outperform.

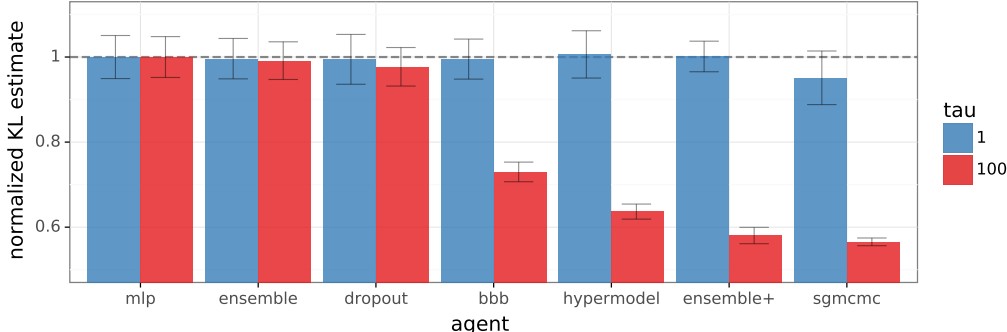

Figure 2: Most Bayesian deep learning approaches do not significantly outperform a single MLP in marginal predictions ($\tau = 1$). Once we examine predictive distributions beyond marginals we see a clear difference in performance between our benchmark agents ($\tau = 100$). For each $\tau$, the KL estimates are normalized by the KL of the MLP agent.

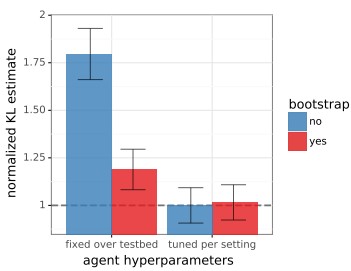

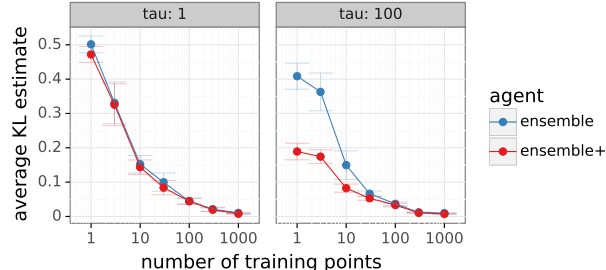

Figure 3: Agent robustness improves with bootstrapping.

Figure 4: The benefits of additive prior functions are clear in the high tau, low data regime.

One of the keys to this result is that all of the agents are able to tune their hyperparameters, such as $L_2$ weight decay, to the SNR regime and number of training points. This matches the way deep learning systems are typically implemented in practice, with extensive hyperparameter tuning on validation data. This methodology has led many practitioners to doubt the usefulness of automatic tuning procedures such as bootstrap sampling (Nixon et al., 2020). In Figure 3, we compare the performance of an `ensemble+` agent that uses bootstrapping with and without the ability to tune the hyperparameters per problem setting. We see that bootstrap sampling is beneficial when the agent is expected to work robustly over a wide range of problem settings. However, the benefits are no longer apparent when the agent is allowed to tune its hyperparameters to individual tasks.

### 4.3 Performance beyond marginals

One of the key contributions of this paper is to evaluate predictive distributions beyond marginals. In Figure 2, the red bars show the results of benchmark agents evaluated on joint predictive distributions with $\tau = 100$. Unlike when evaluating on marginal predictions, where no method significantly outperforms a well-tuned MLP, the potential benefits afforded by Bayesian deep learning become clear when examining higher-order predictive distributions. Our results refute prior works' claims that examining $\mathbf{d}_{\mathrm{KL}}^{\tau}$ beyond marginals provides little new information (Wang et al., 2021).

Figure 2 shows that `sgmcmc` is the top-performing agent overall. This should be reassuring to the Bayesian deep learning community and beyond. In the limit of large compute this agent should recover the 'gold-standard' of Bayesian inference, and it does indeed perform best (Welling & Teh, 2011). However, some of the most popular approaches in this field (`ensemble`, `dropout`) do not actually provide good approximations to the predictive distribution in $\tau = 100$. In fact, we see that even though Bayesian purists may deride `ensemble+` and `hypermodels` as 'not really Bayesian', these methods actually provide much better approximations to the Bayesian posterior than 'fully Bayesian' VI approaches like `bbb`. We

note too that while `sgmcmc` performs best, it also requires orders of magnitude more computation than competitive methods even in this toy setting (see Appendix C.2). As we scale to more complex environments, it may therefore be worthwhile to consider alternative approaches to approximate Bayesian inference.

For insight into where our top agents are able to outperform, we compare `ensemble` and `ensemble+` under the medium SNR regime in Figures 4 and 5. These methods are identical, except for the addition of a randomized prior function (Osband et al., 2018). Figure 4 shows that, although these methods perform similarly in the quality of their marginal predictions ($\tau = 1$), the addition of a prior function greatly improves the quality of joint predictive distributions ($\tau = 100$) in the low data regime. Figure 5 provides additional intuition into *how* the randomized prior functions are able to drive improved performance. Figure 5a shows a sampled generative model from our Testbed, with the training data shown in red and blue circles. Figure 5b shows the mean predictions and 4 randomly sampled ensemble members from each agent (top=`ensemble`, bottom=`ensemble+`). We see that, although the agents mostly agree in their mean predictions, `ensemble+` produces more diverse sampled outcomes enabled by the addition of randomized prior functions. In contrast, `ensemble` produces similar samples, which may explain why its performance is close to baseline `mlp`.

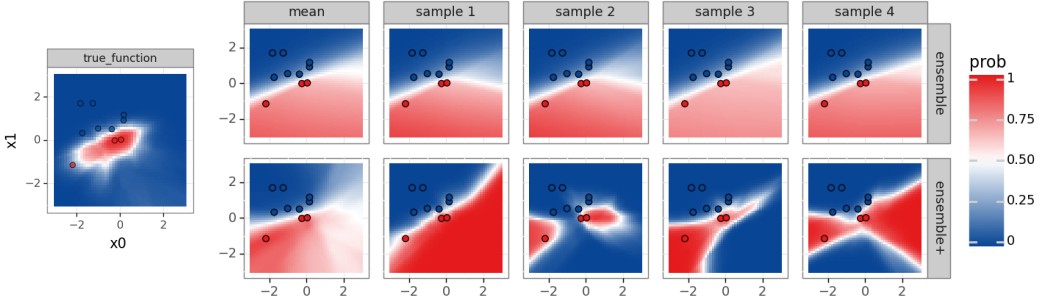

(a) True model.      (b) Agent samples: only ensemble+ produces diverse decision boundaries.

Figure 5: Visualization of the predictions of ensemble and ensemble+ agents.

## 5    Performance on real data

Section 4 provides a simple, sanitized testbed for clear insight to the efficacy of Bayesian deep learning techniques. However, most deep learning research is not driven by these sorts of synthetic generative models, but the ultimate goal of performing well on real datasets. In this section, we apply the same benchmark agents to a selection of small challenge datasets. We find that, on average, tuning agents for the synthetic problems leads to better performance on real data. We also find that, just as the synthetic testbed, agents that perform similarly in marginal predictions may be distinguished in the quality of their joint predictions.

### 5.1    Datasets

We focus on 10 benchmark datasets (3 feature-based, 7 image from pixels) drawn from the literature including Iris, MNIST, and CIFAR-10 (TFD). This collection is not intended to be comprehensive, or to include the most challenging large-scale problems, but instead to represent some canonical real-world data that might reasonably be addressed with the MLP models of Section 4.1. We apply a basic pre-processing step to each dataset, normalizing input features and flattening observations. We push full details to Appendix D.1.

To assess performance in real datasets, we follow a similar procedure as Algorithm 1. The only difference is that since it is impossible to compute the likelihood of environment for real datasets, we compute the negative log-likelihood (NLL) rather than $\mathbf{d}_{KL}^{\tau}$. Appendix A.2 provides further details. Note that NLL and $\mathbf{d}_{KL}^{\tau}$ are equivalent for agent comparison since they differ by a constant (see Equation 1). Furthermore, to allow for more direct comparison with the synthetic testbed, we also consider variants of each dataset where the number of training pairs is limited to less than the 'full' dataset size.

## 5.2 Synthetic data is predictive of real data

Recall that Figure 2 compares performance across an array of agents, assessed using our synthetic data generating process. Each agent's hyperparameters were tuned by first enumerating a list of plausibly near-optimal choices and selecting the one that optimizes performance. Each of our real-world datasets can be viewed as generated by an environment sampled from an alternative data generating process. A natural question is whether performance on the synthetic data correlates with performance on the real-world data.

The table of Figure 6a displays results pertaining to each of our agents. For each agent, performance for each candidate hyperparameter setting was assessed on synthetic and real data, and the correlation across these pairs is reported. The left and right columns restrict attention to datasets with low and high volumes of training data, respectively. If a correlation were equal to 1, the hyperparameter setting that optimizes agent performance on real data would be identical to that on synthetic data. It is reassuring that the correlations are high, reflecting a strong degree of alignment, with the exception of `bbb` in low data regime, for which there appear to be pathological outcomes distorting performance for small training sets. The values in parentheses express 5th and 95th percentile confidence bounds, measured via the statistical bootstrap.

Figure 6b plots performance on real versus synthetic data for the high data regime. Each data point represents one agent-hyperparameter combination. If the correlation were equal to 1, the combination that performs best on the synthetic data would also perform best on the real data. It is reassuring that the correlation is large, and the confidence interval between the 5th and 95th percentiles small. Agent-hyperparameter combinations that perform better on the testbed tend to perform better on real data as well.

| agent | low data | high data |
|---|---|---|
| mlp | 0.74 (0.57,0.85) | 0.68 (0.38,0.99) |
| ensemble | 0.72 (0.52,0.85) | 0.63 (0.34,0.96) |
| dropout | 0.77 (0.68,0.86) | 0.78 (0.66,0.87) |
| bbb | -0.48 (-0.6,-0.35) | 0.76 (0.68,0.83) |
| sgmcmc | 0.72 (0.53,0.85) | 0.86 (0.79,0.92) |
| ensemble+ | 0.85 (0.63,0.98) | 0.74 (0.3,0.97) |
| hypermodel | 0.52 (0.17,0.76) | 0.33 (0.03,0.59) |

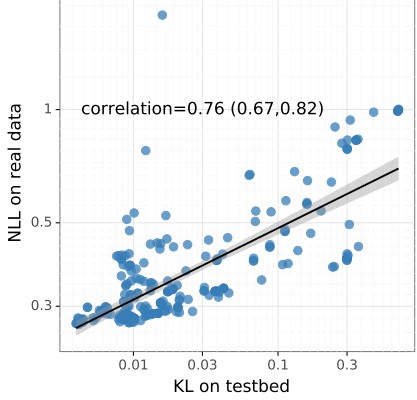

(a) Correlation by agent by data regime.    (b) Correlation in high data regime.

Figure 6: Performance on the Testbed correlates with performance on real datasets.

## 5.3 Higher order predictions and informative priors

Our synthetic testbed can be helpful in driving innovations that carry over to real data. Section 5.2 indicated that performance on the Testbed is correlated with that on real-world data. We now repeat the observation from Figure 4 on real data; additive prior functions can significantly improve the accuracy of joint predictive distributions generated by ensembles. We show this by comparing the performance of `ensemble+` with different forms of prior functions on benchmark datasets. We evaluate an `ensemble` with no prior function (`none`), a random MLP prior (MLP), and a random linear function of a 2-dimensional latent representation as the prior, trained via variational autoencoder (VAE) (Kingma & Welling, 2014). We provide full details in Appendix D.3.

Figure 7 plots the improvement in NLL for the ensemble agent relative to a baseline MLP (lower is better), and breaks out the result for datasets=MNIST,Iris and $\tau = 1, 100$. We

can see that the results for Iris mirror our synthetic data almost exactly. The results for MNIST share some qualitative insights, but also reveal some important differences. For Iris $\tau = 1$ none of the methods outperform the MLP baseline, but for $\tau = 100$ we see significant benefits to an additive MLP prior in the low data regime. For MNIST $\tau = 1$ we actually see benefits to ensembles, even without prior functions and even in the high data regime. This reveals some aspects of this real data that are not captured by our synthetic model, where we did not see this behaviour. For $\tau = 100$ the random MLP prior gives a slight advantage, but the effect is much less pronounced. We hypothesize this is because, unlike the testbed, the MLP prior is not well-matched to the input image data. However, the VAE prior is able to provide significant benefit in the low data regime.[3] These benefits also carry over to Iris, even where random MLPs already provided signficant value. Designing architectures that offer useful priors for learning agents is an exciting area for future work.

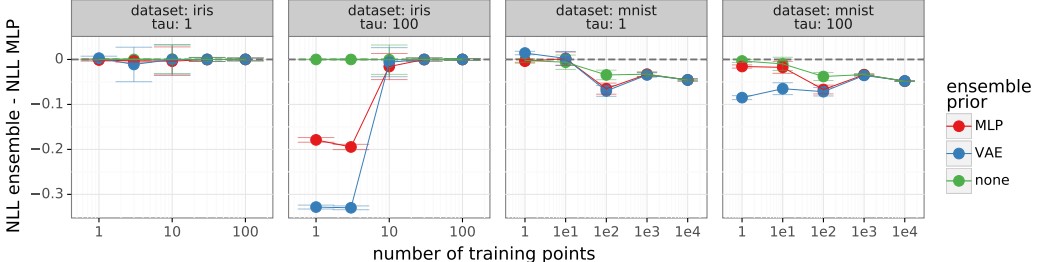

Figure 7: Prior functions provide significant benefit in the high tau, low data regime, just like the testbed. However, for image datasets random MLP priors provide relatively little benefit. Unsupervised pretraining can help to design useful priors in high dimensional data.

## 6    Conclusion

This paper highlights the need to evaluate predictive distributions beyond marginals. In addition to this conceptual contribution, we develop a suite of practical computational tools that can evaluate diverse approaches to uncertainty estimation. Together with these tools, we provide a neural-network-based data generating process that facilitates research and iteration beyond a small set of challenge datasets. We package these together as *The Neural Testbed*, including a variety of baseline agent implementations. We believe that this represents an exciting and valuable new benchmark for Bayesian deep learning and beyond.

We have already used this testbed to generate several new insights in this paper. We have shown many popular Bayesian deep learning approaches perform similarly in marginal predictions but quite differently in joint predictions. We reveal the importance of bootstrapping for parameter robustness, and also help reconcile the observed lack of improvement when tuned to specific datasets. We have shown that these insights from synthetic data can carry over to real datasets; that performance in these settings is correlated, that agents with similar marginal predictions can be distinguished by their joint predictions, and that suitable prior functions can play an important role in driving good performance.

The results in this paper are in some sense preliminary. The grand challenge for Bayesian deep learning is to provide effective uncertainty estimates in large, rich datasets. While we have demonstrated benefits to predictive evaluation beyond marginals only in the 'low data' regime and small-scale problems, we believe that they will extend more broadly to situations where new test inputs appear novel relative to training data. As such, we believe our core insights should carry over to the related problems of nonstationarity and covariate shift that plague modern deep learning systems. As an agent takes on more and more complex tasks, it will continue to run into new and unfamiliar settings and uncertain outcomes; as such, effective predictive distributions will be more important than ever.

---

[3]We hypothesize that appropriately initialized convnet architectures may be able to leverage image structure as noted in prior work (Ulyanov et al., 2018).

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

## A   Testbed Pseudocode

We present the testbed pseudocode in this section. Specifically, Algorithm 2 is the pseudocode for our neural testbed, and Algorithm 3 and Algorithm 4 are two different approaches to estimate the likelihood of a test data $\tau$-sample conditioned on an agent's belief. Algorithm 3 is based on the standard Monte-Carlo estimation, while Algorithm 4 adopts a random partitioning approach. The presented testbed pseudocode works for any prior $\mathbb{P}(\mathcal{E} \in \cdot)$ over the environment and any input distribution $P_X$, including the ones described in Section 4.1. We also release full code and implementations at https://anonymous.4open.science/r/neural-testbed-B839.

In addition to presenting the testbed pseudocode, we also discuss some core technical issues in the neural testbed design. Specifically, Appendix A.1 discusses how to estimate the likelihood of an agent's belief distribution; Appendix A.2 discusses how to extend the testbed to agent evaluation on real data; finally, Appendix A.3 explains our choices of experiment parameters.

---

**Algorithm 2** Neural Testbed

---

**Require:** the testbed requires the following inputs

    1. prior distribution over the environment $\mathbb{P}(\mathcal{E} \in \cdot)$, input distribution $P_X$

    2. agent $f_\theta$

    3. number of training data $T$, test distribution order $\tau$

    4. number of sampled problems $J$, number of test data samples $N$

    5. parameters for agent likelihood estimation, as is specified in Algorithm 3 and 4

**for** $j = 1, 2, \ldots, J$ **do**

    **Step 1: sample environment and training data**

        1. sample environment $\mathcal{E} \sim \mathbb{P}(\mathcal{E} \in \cdot)$

        2. sample $T$ inputs $X_0, X_1, \ldots, X_{T-1}$ i.i.d. from $P_X$

        3. sample the training labels $Y_1, \ldots, Y_T$ conditionally i.i.d. as

$$Y_{t+1} \sim \mathbb{P}\left(Y \in \cdot | \mathcal{E}, X = X_t\right) \quad \forall t = 0, 1, \ldots, T-1$$

        4. choose the training dataset as $\mathcal{D}_T = \{(X_t, Y_{t+1}), t = 0, \ldots, T-1\}$

    **Step 2: train agent**

        train agent $f_{\theta_T}$ based on training dataset $\mathcal{D}_T$

    **Step 3: compute likelihoods**

        **for** $n = 1, 2, \ldots, N$ **do**

            1. sample $X_T^{(n)}, \ldots, X_{T+\tau-1}^{(n)}$ i.i.d. from $P_X$

            2. generate $Y_{T+1}^{(n)}, \ldots, Y_{T+\tau}^{(n)}$ conditionally independently as

$$Y_{t+1}^{(n)} \sim \mathbb{P}\left(Y \in \cdot \Big| \mathcal{E}, X = X_t^{(n)}\right) \quad \forall t = T, T+1, \ldots, T+\tau-1$$

            3. compute the likelihood under the environment $\mathcal{E}$ as

$$p_{j,n} = \mathbb{P}\left(Y_{T+1:T+\tau}^{(n)} \Big| \mathcal{E}, X_{T:T+\tau-1}^{(n)}\right) = \prod_{t=T}^{T+\tau-1} \Pr\left(Y_{t+1}^{(n)} \Big| \mathcal{E}, X_t^{(n)}\right)$$

            4. estimate the likelihood conditioned on the agent's belief

$$\hat{p}_{j,n} \approx \mathbb{P}\left(\hat{Y}_{T+1:T+\tau} = Y_{T+1:T+\tau}^{(n)} \Big| \theta_T, X_{T:T+\tau-1}^{(n)}, Y_{T+1:T+\tau}^{(n)}\right),$$

            based on Algorithm 3 or 4 with test data $\tau$-sample $\left(X_{T:T+\tau-1}^{(n)}, Y_{T+1:T+\tau}^{(n)}\right)$.

    **return** $\frac{1}{JN} \sum_{j=1}^{J} \sum_{n=1}^{N} \log\left(p_{j,n}/\hat{p}_{j,n}\right)$

---

---

**Algorithm 3** Monte Carlo Estimation of Likelihood of Agent's Belief

---

**Require:**
    1. trained agent $f_{\theta_T}$ and number of Monte Carlo samples $M$
    2. test data $\tau$-sample $(X_{T:T+\tau-1}, Y_{T+1:T+\tau})$

**Step 1:** sample $M$ models $\hat{\mathcal{E}}_1, \ldots, \hat{\mathcal{E}}_M$ conditionally i.i.d. from $\mathbb{P}\left(\hat{\mathcal{E}} \in \cdot \Big| f_{\theta_T}\right)$

**Step 2:** estimate $\hat{p}$ as

$$\hat{p} = \frac{1}{M} \sum_{m=1}^{M} \mathbb{P}\left(\hat{Y}_{T+1:T+\tau} = Y_{T+1:T+\tau} \Big| \hat{\mathcal{E}}_m, X_{T:T+\tau-1}, Y_{T+1:T+\tau}\right)$$

**return** $\hat{p}$

---

**Algorithm 4** Estimation of Likelihood of Agent's Belief via Random Partitioning

---

**Require:**
    1. trained agent $f_{\theta_T}$
    2. number of Monte Carlo samples $M$
    3. number of hyperplanes $d$
    4. test data $\tau$-sample $(X_{T:T+\tau-1}, Y_{T+1:T+\tau})$

**Step 1:** sample $M$ models $\hat{\mathcal{E}}_1, \ldots, \hat{\mathcal{E}}_M$ conditionally i.i.d. from $\mathbb{P}(\hat{\mathcal{E}} \in \cdot | f_{\theta_T})$; for each model $m = 1, \ldots, M$, class $k$, and $t = T, \ldots, T+\tau-1$, define

$$p_{m,t,k} = \mathbb{P}(\hat{Y}_{t+1}^{(m)} = k | \hat{\mathcal{E}}_m, X_t),$$

and $\ell_{m,t,k} = \Phi^{-1}(p_{m,t,k})$, where $\Phi(\cdot)$ is the CDF of the standard normal function. For each model $m$, define a vector

$$\ell_m = [\ell_{m,T,1}, \ell_{m,T,2}, \ldots, \ell_{m,T+\tau-1,K}] \in \Re^{K\tau}$$

**Step 2:** sample a $d \times (K\tau)$ matrix $A$ and a $d$-dimensional vector $b$, with each element/component sampled i.i.d. from $N(0,1)$. For each $m = 1, \ldots, M$, compute

$$\psi_m = \mathbf{1}\left[A\ell_m + b \geq 0\right] \in \{0,1\}^d.$$

**Step 3:** partition the sampled models, with each cell indexed by $\psi \in \{0,1\}^d$ and defined by

$$\mathcal{M}_\psi = \{m : \psi_m = \psi\}$$

and assign a probability to each cell:

$$q_\psi = \frac{|\mathcal{M}_\psi|}{M}$$

**Step 4:** $\forall \psi \in \{0,1\}^d$ and $\forall t = T, T+1, \ldots, T+\tau-1$, estimate the probability of predicting $\hat{Y}_{t+1} = k$ conditioned on the cell:

$$p_{\psi,t,k} = \begin{cases} \frac{1}{|\mathcal{M}_\psi|} \sum_{m \in \mathcal{M}_\psi} p_{m,t,k} & \text{if } |\mathcal{M}_\psi| > 0 \\ 1 & \text{if } |\mathcal{M}_\psi| = 0 \end{cases}$$

**Step 5:** estimate $\Pr(\hat{Y}_{t+1:T+\tau} = Y_{t+1:T+\tau} | \theta_T, X_{t:T+\tau-1}, Y_{t+1:T+\tau})$ as

$$\hat{p} = \sum_{\psi \in \{0,1\}^d} q_\psi \prod_{t=T}^{T+\tau-1} p_{\psi,t,Y_{t+1}}$$

**return** $\hat{p}$

---

## A.1 Estimating Likelihood of Agent's Belief Distribution

We have presented two algorithms to estimate the likelihood of a test data $\tau$-sample conditioned on a trained agent: Algorithm 3 is based on the standard Monte Carlo estimation, while Algorithm 4 adopts an approach combining random partitioning and Monte Carlo estimation. In this subsection, we briefly discuss the pros and cons between these two algorithms, and provide some general guidelines on how to choose between them.

Algorithm 3 produces unbiased estimates of the likelihoods, which is usually accurate when $\tau$ is small (e.g. for $\tau \leq 10$). However, maintaining accuracy might require the number of samples $M$ to grow exponentially with $\tau$. The following is an illustrative example.

**Example 1 (Uniform belief over deterministic models):** Consider a scenario where the number of class labels is $K = 2$. We say a model $\hat{\mathcal{E}}$ is deterministic if for any feature vector $X_t$,

$$\mathbb{P}(\hat{Y}_{t+1} = 1 \,|\, \hat{\mathcal{E}}, X_t) \in \{0, 1\}.$$

Obviously, for any test data $\tau$-sample $(X_{T:T+\tau-1}, Y_{T+1:T+\tau})$ with $Y_{T+1:T+\tau} \in \{0,1\}^\tau$, under a deterministic model $\hat{\mathcal{E}}$, we have

$$\mathbb{P}\left(\hat{Y}_{T+1:T+\tau} = Y_{T+1:T+\tau} \,\middle|\, \hat{\mathcal{E}}, X_{T:T+\tau-1}, Y_{T+1:T+\tau}\right) \in \{0, 1\}.$$

When restricted to the inputs $X_{T:T+\tau-1}$, there are $2^\tau$ distinguishable deterministic models. Assume the agent's belief distribution is uniform over these $2^\tau$ distinguishable deterministic models, then for any $Y_{T+1:T+\tau} \in \{0,1\}^\tau$, the likelihood of the agent's belief distribution is

$$\mathbb{P}\left(\hat{Y}_{T+1:T+\tau} = Y_{T+1:T+\tau} \,\middle|\, \theta_T, X_{T:T+\tau-1}, Y_{T+1:T+\tau}\right) = 2^{-\tau}.$$

Now let's consider Algorithm 3. When a model $\hat{\mathcal{E}}_m$ is sampled from the agent's belief distribution, with probability $2^{-\tau}$,

$$\mathbb{P}\left(\hat{Y}_{T+1:T+\tau} = Y_{T+1:T+\tau} \,\middle|\, \hat{\mathcal{E}}_m, X_{T:T+\tau-1}, Y_{T+1:T+\tau}\right) = 1,$$

and with probability $1 - 2^{-\tau}$,

$$\mathbb{P}\left(\hat{Y}_{T+1:T+\tau} = Y_{T+1:T+\tau} \,\middle|\, \hat{\mathcal{E}}_m, X_{T:T+\tau-1}, Y_{T+1:T+\tau}\right) = 0.$$

Consequently, in expectation, we need the number of Monte Carlo samples $M = \Omega(2^\tau)$ to ensure that the estimate $\hat{p}$ returned by Algorithm 3 is non-zero.

To overcome this challenge, we also propose a novel approach to estimate the likelihood of agent's belief via a combination of randomized partitioning and Monte Carlo simulation, as is presented in Algorithm 4. This approach proceeds as follows. First, $M$ models are sampled from the agent's belief distribution. For each sampled model, each test data input $X_t$, and each class label $k$, a predictive probability $p_{m,t,k}$ and its probit $\ell_{m,t,k} = \Phi^{-1}(p_{m,t,k})$ are computed, where $\Phi(\cdot)$ is the CDF of the standard normal distribution. For each sampled model, we also stack its probits into a probit vector $\ell_m \in \Re^{K\tau}$. Then, $d$ random hyperplanes are sampled and used to partition $\Re^{K\tau}$ into $2^d$ cells. Stacked probit vectors place models in cells. Predictive distributions of models in each cell are averaged, and the likelihood is calculated based on these averages, with each cell weighted according to the number of models it contains.

The Neural Testbed applies Algorithm 4 with $2^d \ll M$. Hence, some cells are assigned many models. We conjecture that, under suitable regularity conditions, models assigned to the same cell tend to generate similar predictions. If this is the case, this algorithm produces accurate estimates even when $\tau$ is large. We leave a formal analysis to future work.

Finally, we briefly discuss how to choose between Algorithm 3 and Algorithm 4. As a rule of thumb, we recommend to choose Algorithm 3 for $\tau < 10$ and Algorithm 4 with the number of hyperplanes $d$ between 5 and 10 for $\tau \geq 10$.

## A.2 Agent Evaluation on Real Data

Algorithm 2 (and its simplified version Algorithm 1) is developed for a synthetic data generating processes. We now discuss how to extend it to agent evaluation on real data. Consider a scenario with $J$ real datasets, and each dataset is further partitioned into a training dataset and a test dataset. The main difference between this scenario and a synthetic data generating process is that we cannot compute the likelihood of environment for real data. Thus, we compute the cross-entropy loss instead (see Equation 1). The computational approach is similar to Algorithm 1: for each real dataset, we use its training dataset to train an agent. Then, we sample $N$ test data $\tau$-samples from the test dataset, and estimate the likelihoods of the agent's belief distribution. The estimate of the cross-entropy loss is taken to be the sample mean of the negative log-likelihoods.

Note that when ranking agents, the cross-entropy loss and $\mathbf{d}_{\mathrm{KL}}^{\tau}$ will lead to the same order of agents, since these two losses differ by a constant independent of the agent (see Equation 1).

## A.3 Choices of Experiment Parameters

To apply Algorithm 2, we need to specify an input distribution $P_X$ and a prior distribution on the environment $\mathbb{P}(\mathcal{E} \in \cdot)$. Recall from Section 4.1 that we consider binary classification problems with input dimension 2. We choose $P_X = N(0, I)$, and we consider three environment priors distinguished by a temperature parameter that controls the signal-to-noise ratio (SNR) regime. We sweep over temperatures in $\{0.01, 0.1, 0.5\}$. The prior distribution $\mathbb{P}(\mathcal{E} \in \cdot)$ is induced by a distribution over MLPs with 2 hidden layers and ReLU activation. The MLP is distributed according to standard Xavier initialization, except that biases in the first layer are drawn from $N(0, \frac{1}{2})$. The MLP outputs two units, which are divided by the temperature parameter and passed through the softmax function to produce class probabilities. The implementation of this generative model is in our open source code under the path `/generative/factories.py`.

We now describe the other parameters we use in the Testbed. In Algorithm 2, we pick the order of predictive distributions $\tau \in \{1, 100\}$, training dataset size $T \in \{1, 3, 10, 30, 100, 300, 1000\}$, number of sampled problems $J = 10$, and number of testing data $\tau$-samples $N = 1000$. We apply Algorithm 3 for evaluation of $\mathbf{d}_{\mathrm{KL}}^{1}$ and Algorithm 4 for evaluation of $\mathbf{d}_{\mathrm{KL}}^{100}$. In both Algorithms 3 and 4, we sample $M = 1000$ models from the agent. In Algorithm 4, we set the number of hyperplanes $d = 7$. The specification of the testbed parameters is in our open soucre code under the path `/leaderboard/sweep.py`.

On real datasets, we apply the same $\tau \in \{1, 100\}$, $N = 1000$, and $M = 1000$. We set the number of hyperplanes $d = 10$ in Algorithm 4.

# B  AGENTS

In this section, we describe the benchmark agents in Section 3 and the choice of various hyperparameters used in the implementation of these agents. The list of agents include MLP, ensemble, dropout, Bayes by backprop, stochastic Langevin MCMC, ensemble+ and hypermodel. We will also include other agents such as KNN, random forest, and deep kernel, but the performance of these agents was worse than the other benchmark agents, so we chose not to include them in the comparison in Section 4. In each case, we attempt to match the "canonical" implementation. The complete implementation of these agents including the hyperparameter sweeps used for the Testbed are available at https://anonymous.4open.science/r/neural-testbed-B839. We make use of the Epistemic Neural Networks notation from (Osband et al., 2021) in our code. We set the default hyperparameters of each agent to be the ones that minimize the aggregated KL score $\mathbf{d}_{\mathrm{KL}}^{\mathrm{agg}} = \mathbf{d}_{\mathrm{KL}}^{1} + \mathbf{d}_{\mathrm{KL}}^{100}/100$.

## B.1  MLP

The `mlp` agent learns a 2-layer MLP with 50 hidden units in each layer by minimizing the cross-entropy loss with $L_2$ weight regularization. The $L_2$ weight decay scale is chosen either to be $\lambda\frac{1}{T}$ or $\lambda\frac{d\sqrt{\beta}}{T}$, where $d$ is the input dimension, $\beta$ is the temperature of the generative process and $T$ is the size of the training dataset. We sweep over $\lambda \in \{10^{-4}, 10^{-3}, 10^{-2}, 10^{-1}, 1, 10, 100\}$. We implement the MLP agent as a special case of a deep ensemble (B.2). The implementation and hyperparameter sweeps for the `mlp` agent can be found in our open source code, as a special case of the `ensemble` agent, under the path `/agents/factories/ensemble.py`.

## B.2  ENSEMBLE

We implement the basic "deep ensembles" approach for posterior approximation (Lakshminarayanan et al., 2017). The `ensemble` agent learns an ensemble of MLPs by minimizing the cross-entropy loss with $L_2$ weight regularization. The only difference between the ensemble members is their independently initialized network weights. We chose the $L_2$ weight scale to be either $\lambda\frac{1}{MT}$ or $\lambda\frac{d\sqrt{\beta}}{MT}$, where $M$ is the ensemble size, $d$ is the input dimension, $\beta$ is the temperature of the generative process, and $T$ is the size of the training dataset. We sweep over ensemble size $M \in \{1, 3, 10, 30, 100\}$ and $\lambda \in \{10^{-4}, 10^{-3}, 10^{-2}, 10^{-1}, 1, 10, 100\}$. We find that larger ensembles work better, but this effect is within margin of error after 10 elements. The implementation and hyperparameter sweeps for the `ensemble` agent can be found in our open source code under the path `/agents/factories/ensemble.py`.

## B.3  DROPOUT

We follow Gal & Ghahramani (2016) to build a `dropout` agent for posterior approximation. The agent applies dropout on each layer of a fully connected MLP with ReLU activation and optimizes the network using the cross-entropy loss combined with $L_2$ weight decay. The $L_2$ weight decay scale is chosen to be either $\frac{l^2}{2T}(1 - p_{\mathrm{drop}})$ or $\frac{d\sqrt{\beta}l}{T}$ where $p_{\mathrm{drop}}$ is the dropping probability, $d$ is the input dimension, $\beta$ is the temperature of the data generating process, and $T$ is the size of the training dataset. We sweep over dropout rate $p_{\mathrm{drop}} \in \{0.1, 0.2, 0.3, 0.4, 0.5, 0.6, 0.7, 0.8\}$, length scale (used for $L_2$ weight decay) $l \in \{0.01, 0.1, 0.3, 1, 3, 10\}$, number of neural network layers $\in \{2, 3\}$, and hidden layer size $\in \{50, 100\}$. The implementation and hyperparameter sweeps for the `dropout` agent can be found in our open source code under the path `/agents/factories/dropout.py`.

## B.4  BAYES-BY-BACKPROP

We follow Blundell et al. (2015) to build a `bbb` agent for posterior approximation. We consider a scale mixture of two zero-mean Gaussian densities as the prior. The Gaussian densities have standard deviations $\sigma_1$ and $\sigma_2$, and they are mixed with probabilities $p$ and $1 - p$,

respectively. We sweep over $\sigma_1 \in \{1, 2, 4\}$, $\sigma_2 \in \{0.25, 0.5, 0.75\}$, $p \in \{0, 0.25, 0.5, 0.75, 1\}$, learning rate $\in \{10^{-3}, 3 \times 10^{-3}\}$, number of training steps $\in \{500, 1000, 10000\}$, number of neural network layers $\in \{2, 3\}$, hidden layer size $\in \{50, 100\}$, and the ratio of the complexity cost to the likelihood cost $\in \{1, d\sqrt{\beta}\}$, where $d$ is the input dimension and $\beta$ is the temperature of the data generating process. The implementation and hyperparameter sweeps for the `bbb` agent can be found in our open source code under the path `/agents/factories/bbb.py`.

## B.5 Stochastic gradient Langevin dynamics

We follow Welling & Teh (2011) to implement a `sgmcmc` agent using stochastic gradient Langevin dynamics (SGLD). We consider two versions of SGLD, one with momentum and other without the momentum. We consider independent Gaussian prior on the neural network parameters where the prior variance is set to be

$$\sigma^2 = \lambda \frac{T}{d\beta},$$

where $\lambda$ is a hyperparameter that is swept over $\{0.01, 0.1, 0.5, 1\}$, $d$ is the input dimension, $\beta$ is the temperature of the data generating process, and $T$ is the size of the training dataset. We consider a constant learning rate that is swept over $\{10^{-5}, 5 \times 10^{-5}, 10^{-4}, 5 \times 10^{-4}, 10^{-3}, 5 \times 10^{-3}, 10^{-2}\}$. For SGLD with momentum, the momentum decay term is always set to be 0.9. The number of training batches is $5 \times 10^5$ with burn-in time of $10^5$ training batches. We save a model every 1000 steps after the burn-in time and use these models as an ensemble during the evaluation. The implementation and hyperparameter sweeps for the `sgmcmc` agent can be found in our open source code under the path `/agents/factories/sgmcmc.py`.

## B.6 Ensemble+

We implement the `ensemble+` agent using deep ensembles with randomized prior functions (Osband et al., 2018) and bootstrap sampling (Osband & Van Roy, 2015). Similar to the vanilla ensemble agent in Section B.2, we consider $L_2$ weight scale to be either $\lambda \frac{1}{MT}$ or $\lambda \frac{d\sqrt{\beta}}{MT}$. We sweep over ensemble size $M \in \{1, 3, 10, 30, 100\}$ and $\lambda \in \{10^{-4}, 10^{-3}, 10^{-2}, 10^{-1}, 1, 10, 100\}$. The randomized prior functions are sampled exactly from the data generating process, and we sweep over prior scaling $\in \{0, \sqrt{\beta}, 1\}$. In addition, we sweep over bootstrap type (none, exponential, bernoulli). We find that the addition of randomized prior functions is crucial for improvement in performance over vanilla deep ensembles in terms of the quality of joint predictions. We also find that bootstrap sampling improves agent robustness, although the advantage is less apparent when one is allowed to tune the $L_2$ weight decay for each task (see Figure 3). The implementation and hyperparameter sweeps for the `ensemble+` agent can be found in our open source code under the path `/agents/factories/ensemble_plus.py`.

## B.7 Hypermodel

We follow Dwaracherla et al. (2020) to build a `hypermodel` agent for posterior approximation. We consider a linear hypermodel over a 2-layer MLP base model. We sweep over index dimension $\in \{1, 3, 5, 7\}$. The $L_2$ weight decay is chosen to be either $\lambda \frac{1}{T}$ or $\lambda \frac{d\sqrt{\beta}}{T}$ with $\lambda \in \{0.1, 0.3, 1, 3, 10\}$, where $d$ is the input dimension, $\beta$ is the temperature of the data generating process, and $T$ is the size of the training dataset. We chose three different bootstrapping methods of none, exponential, bernoulli. We use an additive prior which is a linear hypermodel prior over an MLP base model, which is similar to the generating process, with number of hidden layers in $\{1, 2\}$, 10 hidden units in each layer, and prior scale from $\{0, \sqrt{\beta}, 1\}$. The implementation and hyperparameter sweeps for the `hypermodel` agent can be found in our open source code under the path `/agents/factories/hypermodel.py`.

## B.8 Non-parametric classifiers

K-nearest neighbors (k-NN) (Cover & Hart, 1967) and random forest classifiers (Friedman, 2017) are simple and cheap off-the-shelf non-parametric baselines (Murphy, 2012; Pedregosa et al., 2011). The 'uncertainty' in these classifiers arises merely from the fact that they produce distributions over the labels and as such we do not expect them to perform well relative to more principled approaches. Moreover, these methods have no capacity to model $\mathbf{d}_{\mathrm{KL}}^{\tau}$ for $\tau > 1$. For the `knn` agent we swept over the number of neighbors $k \in \{1, 5, 10, 30, 50, 100\}$ and the weighting of the contribution of each neighbor as either uniform or based on distance. For the `random_forest` agent we swept over the number of trees in the forest $\{10, 100, 1000\}$, and the splitting criterion which was either the Gini impurity coefficient or the information gain. To prevent infinite values in the KL we truncate the probabilities produced by these classifiers to be in the interval $[0.01, 0.99]$. The implementation and hyperparameter sweeps for the `knn` and `random_forest` agents can be found in our open source code under the paths `/agents/factories/knn.py` and `/agents/factories/random_forest.py`.

## B.9 Gaussian process with learned kernel

A neural network takes input $X_t \in \mathcal{X}$ and produces output $Z_{t+1} = W\phi_\theta(X_t) + b \in \mathbf{R}^K$, where $W \in \mathbf{R}^{K \times m}$ is a matrix, $b \in \mathbf{R}^K$ is a bias vector, and $\phi_\theta : \mathcal{X} \to \mathbf{R}^m$ is the output of the penultimate layer of the neural network. In the case of classification the output $Z_{t+1}$ corresponds to the logits over the class labels, *i.e.*, $\hat{Y}_{t+1} \propto \exp(Z_{t+1})$. The neural network should learn a function that maps the input into a space where the classes are linearly distinguishable. In other words, the mapping that the neural network is learning can be considered a form of *kernel* (Schölkopf & Smola, 2018), where the kernel function $k : \mathcal{X} \times \mathcal{X} \to \mathbf{R}$ is simply $k(X, X') = \phi_\theta(X)^\top \phi_\theta(X')$. With this in mind, we can take a *trained* neural network and consider the learned mapping to be the kernel in a Gaussian process (GP) (Rasmussen, 2003), from which we can obtain approximate uncertainty estimates. Concretely, let $\Phi_{0:T-1} \in \mathbf{R}^{T \times m}$ be the matrix corresponding to the $\phi_\theta(X_t)$, $t = 0, \dots, T-1$, vectors stacked row-wise and let $\Phi_{T:T+\tau-1} \in \mathbf{R}^{\tau \times m}$ denote the same quantity for the test set. Fix index $i \in \{0, \dots, K-1\}$ to be a particular class index. A GP models the joint distribution over the dataset to be a multi-variate Gaussian, *i.e.*,

$$\begin{bmatrix} Z_{1:T}^{(i)} \\ Z_{T+1:T+\tau}^{(i)} \end{bmatrix} \sim \mathcal{N}\left( \begin{bmatrix} \mu_{1:T}^{(i)} \\ \mu_{T+1:T+\tau}^{(i)} \end{bmatrix}, \begin{bmatrix} \sigma^2 I + \Phi_{0:T-1}\Phi_{0:T-1}^\top & \Phi_{T:T+\tau-1}\Phi_{0:T-1}^\top \\ \Phi_{0:T-1}\Phi_{T:T+\tau-1}^\top & \Phi_{T:T+\tau-1}\Phi_{T:T+\tau-1}^\top \end{bmatrix} \right)$$

where $\sigma > 0$ models the noise in the training data measurement and $\mu_{1:T}^{(i)}$, $\mu_{T+1:T+\tau}^{(i)}$ are the means under the GP. The conditional distribution is given by

$$P(Z_{T+1:T+\tau}^{(i)} \mid Z_{1:T}^{(i)}, X_{0:T+\tau-1}) = \mathcal{N}\left( \mu_{T+1:T+\tau|1:T}^{(i)}, \Sigma_{T+1:T+\tau|1:T} \right)$$

where

$$\Sigma_{T+1:T+\tau|1:T} = \Phi_{T:T+\tau-1}\Phi_{T:T+\tau-1}^\top - \Phi_{T:T+\tau-1}\Phi_{0:T-1}^\top(\sigma^2 I + \Phi_{0:T-1}\Phi_{0:T-1}^\top)^{-1}\Phi_{0:T-1}\Phi_{T:T+\tau-1}^\top.$$

and rather than use the GP to compute $\mu_{T+1:T+\tau|0:T}^{(i)}$ (which would not be possible since we do not oberve the true logits) we just take it to be the output of the neural network when evaluated on the test dataset. The matrix being inverted in the expression for $\Sigma_{T+1:T+\tau|0:T}$ has dimension $T \times T$, which may be quite large. We use the Sherman-Morrison-Woodbury identity to rewrite it as follows (Woodbury, 1950)

$$\Sigma_{T+1:T+\tau|0:T} = \Phi_{T:T+\tau-1}(I - \Phi_{0:T-1}^\top(\sigma^2 I + \Phi_{0:T-1}\Phi_{0:T-1}^\top)^{-1}\Phi_{0:T-1})\Phi_{T:T+\tau-1}^\top$$
$$= \sigma^2 \Phi_{T:T+\tau-1}(\sigma^2 I + \Phi_{0:T-1}^\top\Phi_{0:T-1})^{-1}\Phi_{T:T+\tau-1}^\top,$$

which instead involves the inverse of an $m \times m$ matrix, which may be much smaller. If we perform a Cholesky factorization of positive definite matrix $(\sigma^2 I + \Phi_{0:T-1}^\top\Phi_{0:T-1}) = LL^\top$ then the samples for all logits simultaneously can be drawn by first sampling $\zeta \in \mathbf{R}^{m \times K}$, with each entry drawn IID from $\mathcal{N}(0, 1)$, then forming

$$\hat{Y}_{T+1:T+\tau} \propto \exp(\mu_{T+1:T+\tau|1:T} + \sigma\Phi_{T:T+\tau-1}L^{-\top}\zeta).$$

The implementation and hyperparameter sweeps for the `deep_kernel` agent can be found in our open source code under the path `/agents/factories/deep_kernel.py`.

## B.10 Other agents

In our paper we have made a concerted effort to include representative and canonical agents across different families of Bayesian deep learning and adjacent research. In addition to these implementations, we performed extensive tuning to make sure that each agent was given a fair shot. However, with the proliferation of research in this area, it was not possible for us to evaluate all competing approaches. We hope that, by opensourcing the Neural Testbed, we can allow researchers in the field to easily assess and compare their agents to these baselines.

For example, we highlight a few recent pieces of research that might be interesting to evaluate in our setting. Of course, there are many more methods to compare and benchmark. We leave this open as an exciting area for future research.

- **Neural Tangent Kernel Prior Functions** (He et al., 2020). Proposes a specific type of prior function in *ensemble+* inspired by connections to the neural tangent kernel.
- **Functional Variational Bayesian Neural Networks** (Sun et al., 2019). Applies variational inference directly to the function outputs, rather than weights like `bbb`.
- **Variational normalizing flows** (Rezende & Mohamed, 2015). Applies variational inference over a more expressive family than `bbb`.
- **No U-Turn Sampler** (Hoffman et al., 2014). Another approach to `sgmcmc` that attempts to compute the posterior directly, computational costs can grow large.

## C Testbed results

In this section, we provide the complete results of the performance of benchmark agents on the Testbed, broken down by the temperature setting, which controls the SNR, and the size of the training dataset. We select the best performing agent within each agent family and plot $\mathbf{d}_{\mathrm{KL}}^{1}$ and $\mathbf{d}_{\mathrm{KL}}^{100}$ with the performance of an MLP agent as a reference. We also provide a plot comparing the training time of different agents.

### C.1 Performance breakdown

Figures 8 and 9 show the KL estimates evaluated on $\tau = 1$ and $\tau = 100$, respectively. For each agent, for each SNR regime, for each number of training points we plot the average KL estimate from the Testbed. In each plot, we include the "baseline" `mlp` agent as a black dashed line to allow for easy comparison across agents. A detailed description of these benchmark agents can be found in Appendix B.

### C.2 Training time

Figure 10 shows a plot comparing the $\mathbf{d}_{\mathrm{KL}}^{100}$ and training time of different agents normalized with that of an MLP. We can see that `sgmcmc` agent has the best performance, but at the cost of more training time (computation). Both `ensemble+` and `hypermodel` agents have similar performance as `sgmcmc` with lower training time. We trained our agents on CPU only systems.

## D Real data

This section provides supplementary details regarding the experiments in Section 5. As before, we include full implementation and source code at https://anonymous.4open.science/r/neural-testbed-B839.

### D.1 Datasets

Table 2 outlines the datasets included in our experiments. Unlike to the synthetic testbed, which evaluates agents over a range of SNR regimes, these datasets are generally all high

SNR regime. We can see this since the top-performing agents in the literature are able to obtain high levels of classification accuracy on held out data; something that is impossible if the underlying system has high levels of noise.

| dataset name | type | # classes | input dimension | # training pairs |
|---|---|---|---|---|
| iris | structured | 3 | 4 | 120 |
| wine quality | structured | 11 | 11 | 3,918 |
| german credit numeric | structured | 2 | 24 | 800 |
| mnist | image | 10 | 784 | 60,000 |
| fashion-mnist | image | 10 | 784 | 60,000 |
| mnist-corrupted/shot-noise | image | 10 | 784 | 60,000 |
| emnist/letters | image | 37 | 784 | 88,800 |
| emnist/digits | image | 10 | 784 | 240,000 |
| cmaterdb | image | 10 | 3,072 | 5,000 |
| cifar10 | image | 10 | 3,072 | 50,000 |

Table 2: Summary of benchmark datasets used in the paper.

Each of these datasets is provided with a canonical training/test set of specific sizes. In order to examine performance in different data regimes we augment the default settings of Table 2 by also examining the performance of agents on these datasets with reduced training data. In a way that mirrors the testbed sweep of Section 4.1, we also look at settings where the training data is restricted to $T = 1, 10, 100, 1000, 10000$ data points respectively.

## D.2 CORRELATION

Figure 6 breaks down the correlation in performance between testbeds and real data. For the purposes of Table 6a we say that $T = 1, 10$ is the 'low data' regime, and the maximum training dataset size is the 'high data' regime. Our results show that, for each agent, for each data regime, performance of hyperparameters is correlated across settings.

One concern might be that while performance on real data *overall* is highly correlated, that this might not necessarily be the case for any individual dataset. Or, alternatively, that this correlation is driven by extremely strong relationships in one dataset that are not present in others. Figure 11 shows that this is not the case. In fact, for each of the datasets considered we have strong and positive correlation over agent-hyperparameter pairs. This gives us confidence that the results of Figure 6b are robust not only to choice of agent, but also to some reasonable choice of datasets.

## D.3 PRIOR FUNCTIONS

We consider two different forms of prior functions for `ensemble+`: a random MLP of the input data and a random linear function of a 2-dimensional latent trained via variational autoencoder (VAE) (Kingma & Welling, 2014). For the MLP prior, we tried both linear (MLP with no hidden layer) and MLP with hidden layers, and observed that the linear prior works better. To train the 2-dimensional latent, we considered a 2-layer (128, 64) MLP for the Gaussian encoder and a 2-layer (64, 128) MLP for the Bernoulli decoder. We trained the VAE using all unsupervised training data available for each dataset. After training the VAE for 10,000 steps, we used the output mean of the Gaussian encoder as the latent.

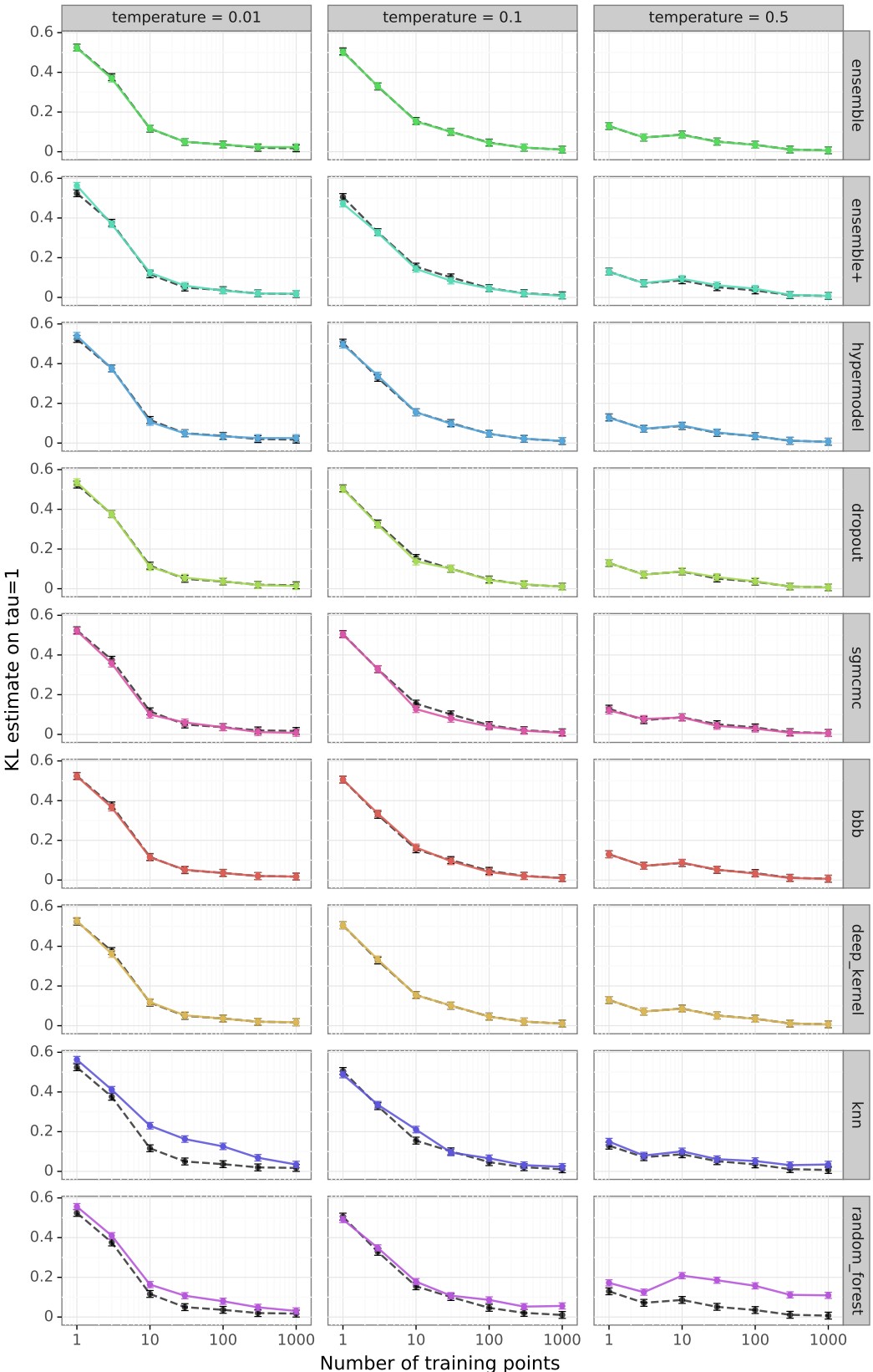

Figure 8: Performance of benchmark agents on the Testbed evaluated on $\tau = 1$, compared against the MLP baseline.

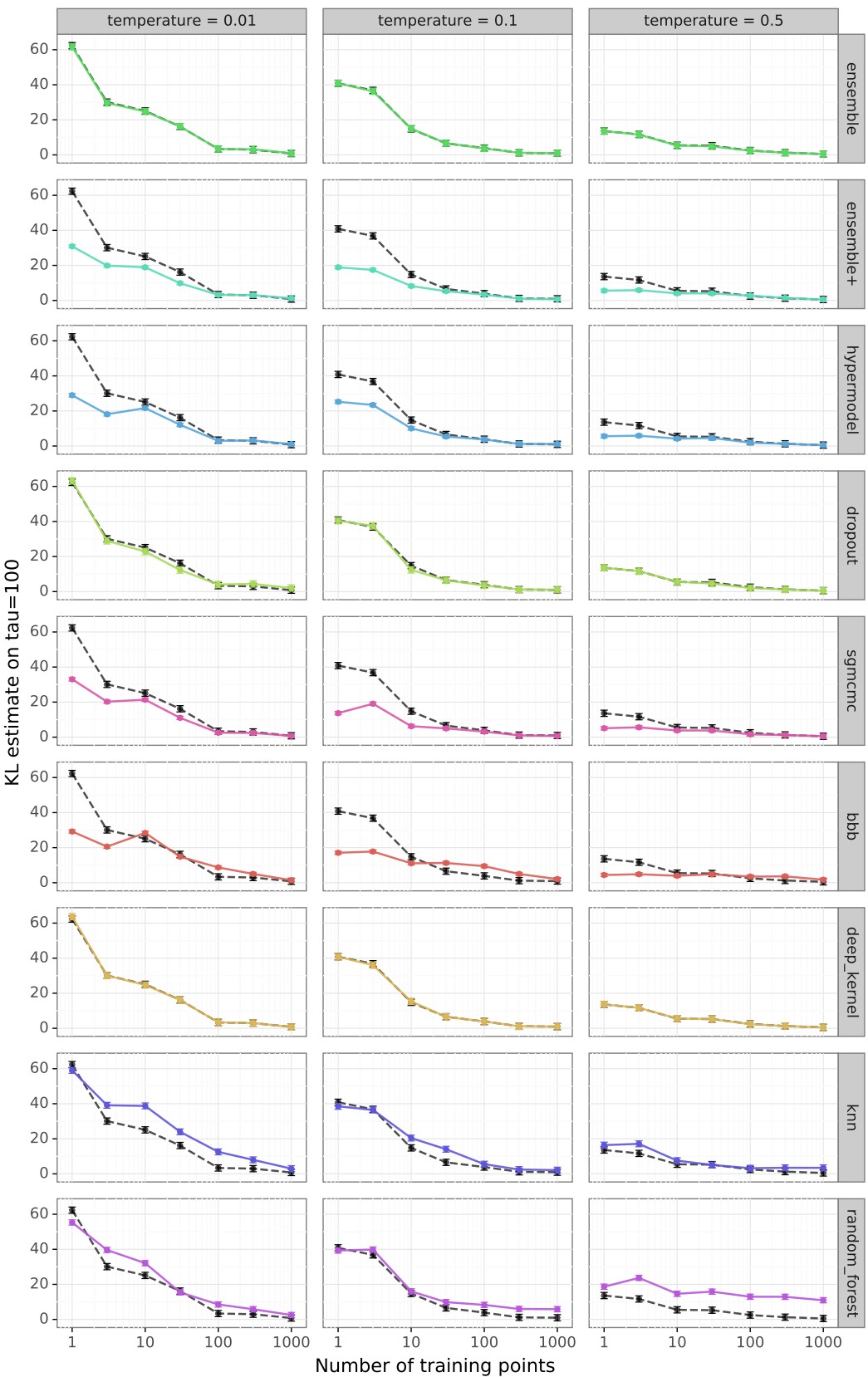

Figure 9: Performance of benchmark agents on the Testbed evaluated on $\tau = 100$, compared against the MLP baseline.

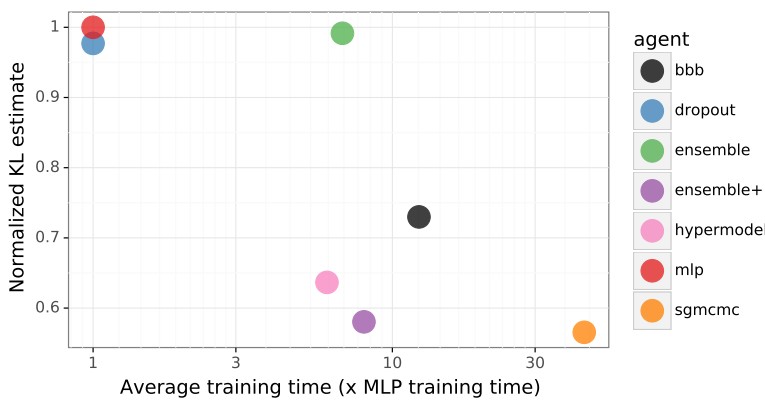

Figure 10: Normalized KL vs training time of different agents

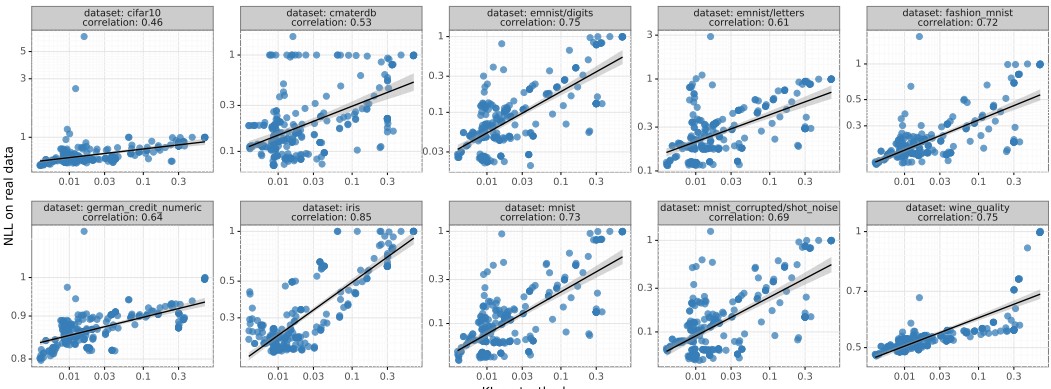

Figure 11: Correlation in high data regime for different datasets.

