# OpenReview forum: "Evaluating Predictive Distributions: Does Bayesian Deep Learning Work?"
_ICLR.cc/2022/Conference — ICLR 2022 Submitted_

### Official Review · Reviewer_E6jF · 2021-10-27

**Correctness:** 4
**Technical Novelty And Significance:** 3
**Empirical Novelty And Significance:** 3
**Recommendation:** 6
**Confidence:** 4

**Main Review:**

_Edit: Thank you for the rebuttal and detailed discussion. I consider the results an interesting contribution that is relevant to the field, which is why I vote for acceptance. The limited score is due to the presentation and discussion of topics in the paper not directly related to the theoretical contribution (see discussion below for details). A proper rewrite of the related sections to guide the reader to focus on the main topic marginal vs joint posterior predictive would strengthen the paper a lot._

___________

## Strengths
- The paper is overall well written and structured
- The experiments demonstrate the difference between marginal and joint posterior predictive the authors ask the community to focus on
- An extensive library with code is provided, which allows for the implementation and test of further methods

## Weaknesses
- The title reads a little bit too attention-grabbing, especially given that the paper itself does not actually target the question of whether Bayesian deep learning (BDL) works, but rather whether evaluating a marginal posterior predictive is sufficient, of whether one should consider joint posterior predictive.
- The abstract claims to solve "what roles epistemic versus aleatoric uncertainty play", and the end of the introduction also claims to "resolve a point of philosophical debate" on this topic. However, can the authors comment to in which respect they do that? As far as I read it, the authors claim that epistemic and aleatoric uncertainty are highly dependent on the model (e.g. they differ depending on the model, yet can lead to the same predictive distributions for distinct models), and that to compare models, one should rely on the posterior predictive. (To be more precise in the rest of the paper, the joint posterior predictive.) But is that really a point that is being debated and needs to be resolved? That what is reducible and irreducible uncertainty is always conditioned on the model, and that we, therefore, need to focus on the posterior predictive to compare different models with each other properly seems to be completely obvious and not challenged by anybody to my knowledge. (Unfortunately, the supposed philosophical debate lacks all references.)
    The realization that we cannot usefully compare ir/reducible uncertainties between models does, of course, not at all tackle the question of whether it can be useful to distinguish between them within, i.e. conditioned on a specific model, to gain a deeper understanding of why the current model performs as it does, or how one can improve it (taking e.g. Depeweg et al. (2018) as an example of an active learning context).
- Section 2.1 and 2.2 read as if they are due to the paper. However, both the $\tau$th-order KL-loss as well as a minor variation of the example are due to Lu et al. (2021), which are only cited in passing without highlighting this fact.
- Given the small nature of the nets further comparisons against an HMC giving access to  the _true_ posterior seems feasible and relevant in the set of models discussed, and if it is infeasible after all an SGHMC (Chen et al., 2014) could give a second point similar to the Langevin `sgmcmc` to have a richer exploration of the posterior.
- Given the topic, Izmailov et al. (2021) seem highly relevant with similar results concerning ensembles and `sgmcmc` performances (abbreviated SGLD in their case). This prior work is missing completely in the discussion.


## Specific additional questions to the authors
- Can the authors clarify the differences/similarities with Lu et al., 2021?
- Can the authors comment in greater detail on the claimed different observations between the current work and Wang et al. (2021)?
- The nets used in the experiments are rather small. Do the authors have any indication that the findings remain stable with deeper nets?


## Minor
- Several parts of the paper are "confusing", "puzzling", "surprising", unclear as to what is "really Bayesian", and seem to be formulated with the primary goal of teasing "Bayesian purists". While this is a nice structure for a poster/conference talk to provoke some good discussions in the next break, it feels somewhat unnecessary in a paper. _(Feel free to ignore this complaint, as it is only my highly subjective prior)_
- Similar to the comment on the teasing structure of the paper and the earlier comments on aleatoric/epistemic, the XKCD comic seems somewhat out of place. Especially given that it is introduced as if the deep learning community is its cause. While I do not know the original story behind Munroe's comic, the discussion of different types of sources of uncertainty is a lot older than our current deep learning popularization.
- "We see that ensemble ... actually provide much better approximations to the Bayesian posterior than 'fully Bayesian' VI approaches like `bbb`". I would claim that this finding is not too surprising. Given that most VI approaches take a unimodal, mean-field approximation as their starting point, it is directly clear that the approximation to a highly multimodal posterior is terrible. It seems reasonable that an ensemble approach gives a richer signal as long as its members do not collapse to the same optimum. This has also been extensively evaluated and demonstrated by Izmailov et al. (2021).
- ~~I might have overlooked it, but the hyperparameter details in the appendix seem to miss the number of samples from the posterior for the BNNs.~~ _Edit: All details are provided, I had just overlooked them. See the author response below_

### Typos
- The datasets in Sec 5.1 give (TFD) as a reference, while the references only contain the long-form TensorFlow Datasets.



______
Depeweg et al. Decomposition of Uncertainty in Bayesian Deep Learning for Efficient and Risk-sensitive Learning, ICML 2018
Izmailov et al. What Are Bayesian Neural Network Posteriors Really Like?, ICML 2021

**Summary Of The Paper:**

The authors discuss whether it is sufficient to consider the marginal posterior predictive vs considering a joint posterior predictive when evaluating Bayesian deep learning approaches. Introducing a set of experiments (_The Neural Testbed_), they demonstrate that the performance of common approaches can differ greatly depending on which of these predictive distributions are evaluated. Additionally, extensive code is provided for efficient implementation and evaluation of new models.


**Summary Of The Review:**

An overall interesting paper in a new direction (after the contribution of Lu et al. is clarified) to evaluate BDL models that tries a bit too hard to be controversial.

---

> ### Author Response · Authors · 2021-11-16
> **Author response**
>
> Thank you very much for your review.
> We are happy that you have identified key strengths of the paper, notably presentation/writing, experimental clarity and open source code.
>
> We have collected our main responses to a shared rebuttal that addresses the main points common to our reviews.
> In particular, we believe that (1), (2), (3) and (5) are all particularly relevant to the discussions raised in the your review.
> We hope that these can help assuage your concerns, and help to improve our submission.
>
> To handle some of the specific comments not handled above:
>
> - We agree the title is a little provocative... I think the idea was "does Bayesian deep learning work *at producing good predictions*" and really highlighting that while many methods perform well at marginals, some other popular approaches are surprisingly terrible at making joint predictions. I don't think this is well-known in the community, where dropout/ensembles are generally viewed as some of the top-performers... but perform terribly in our testbed tau=100.
>
> - Lu et al introduce the tau-th order KL divergence in an arXiv technical note/pre-print, together with some analyis of the importance and higher-order predictions in decision making. Our paper takes this same metric and focuses on the problems of Bayesian deep learning, computing and benchmarking agent performance on these tasks for a conference submission. One might imagine larger journal-style work that combines this line of work for a wholistic view of learning, uncertainty and prediction beyond marginals. We can work to clarify these distinctions and make any novelty/dependencies more clear.
>
> - All BNNs (and other models) are evaluated with M=1000 samples from the posterior, listed in Appendix A.3 and https://anonymous.4open.science/r/neural-testbed-B839/neural_testbed/leaderboard/sweep.py as `num_enn_samples`.
>
> Overall, we believe that you do appreciate the potential value that this paper could offer to stimulate the community... even if it's clear that at times our writing/tone has maybe made this less appealing to you.
> We hope that, in the process of this review we can convince you to become an "advocate" for this paper... particularly since the overall scores are very much borderline.
>
> Many thanks.

---

> > ### Comment · Reviewer_E6jF · 2021-11-17
> > **Response to the author response**
> >
> > Thank you for your response.
> >
> >
> > > All BNNs (and other models) are evaluated with M=1000 samples from the posterior, listed in Appendix A.3 and https://anonymous.4open.science/r/neural-testbed-B839/neural_testbed/leaderboard/sweep.py as num_enn_samples.
> >
> >
> > Thanks for the pointer to code/appendix. I had overlooked the corresponding part.
> >
> > My main problem with the paper lies primarily in the discrepancy between what it delivers and what it claims to deliver.
> >
> > It delivers some interesting and important data on whether one can rely on the marginal predictive for model evaluations or whether one should use the joint. Here, it informs the community that these are not as interchangeable as one might think at first glance.
> > To keep the attention grabbing question structure of the title, a more justified title might be to go in the direction of: "Evaluating Predictive Distributions: Should BDL evaluation switch from the marginal to the joint?"
> >
> > What it claims in the title, however, is to evaluate whether BDL works at all. That would put it into the tradition of e.g. Ovadia et al. (2019), Wenzel et al. (2020), Izmailov et al. (2021). But its actual message, as stated above, is primarily that one should not just rely on the marginal when one tackles such questions, not whether BDL in general is a working approach.
> > The second claim is the epistemic/aleatoric uncertainty discussion. I agree with your point (see (2) in the general author response above) that the distinction is always model dependent, but I do not see what additional insight you provide, given that that fact is nothing new. The provided example shows, in my opinion, why it is important to evaluate the joint vs the marginal posterior predictive, as the underlying uncertainty structure different models assign internally can have a practical influence that the common marginal evaluation does not discover/take into account.
> > But in which sense does that give an insight in _"the importance of epistemic vs aleatoric uncertainty"_? It gives insight into the importance of marginal vs joint predictive evaluation and shows that differences in these two can, for example, be due to underlying differences in the ir/reducible uncertainty structure within the compared models, but I do not see that that is, in general, an equivalence relation as you claim in your answer above. If it were, that would be indeed a strong result that then deserved to be prominently stated in the paper as well.
> >
> >
> > Essentially, I would be happy if the paper followed its own one-sentence openreview summary
> > > This paper introduces The Neural Testbed, which evaluates posterior predictive distributions beyond marginals.
> >
> > in both the abstract and the content, leaving epistemic/aleatoric discussions as more of a small side comment. However, that would probably require a larger rewrite that I cannot advocate for blindly without seeing it beforehand.

---

> > > ### Author Response · Authors · 2021-11-17
> > > **Failure to deliver**
> > >
> > > Thank you for your continued comments and engagement.
> > >
> > > I think that you can definitely make a case that we don't draw a line under absolutely every point raised in our paper.
> > > In hindsight it might have been a better strategy to really focus on just one of the many insights we raise, in order to really to answer that definitively.
> > > However, in doing this, I think we would have ended up with a piece of work that did not stimulate as much future work in the community.
> > > What I want to do here is make a clear case that the paper does deliver on some of these key claims.
> > >
> > > > Does Bayesian Deep Learning Work?
> > >
> > > We follow the tradition of other Bayesian deep learning competitions in examining whether a method "works" by examining the quality its predictive distribution: https://izmailovpavel.github.io/neurips_bdl_competition/.
> > > We provide an extensive evaluation of benchmark methods and tuning in a simple generative model.
> > > Importantly, this comparison sidesteps the morass of whether a method is "really Bayesian", but is compatible with the usual notions of Bayes-optimality.
> > >
> > > Now, unlike previous work, we *also* investigate the quality of joint predictions, as well as marginals.
> > > When we do this, we find **extremely clear separation** of methods into those that work better than a baseline MLP and those that do not...
> > > This is novel in the field, which typically focuses on small percentage tweaks in benchmark datasets, and which are easily conflated with other training tricks such as specific learning rate schedules.
> > >
> > > In particular, our paper helps to clarify whether Dropout "works" as Bayesian approximation... something that has been a consistent topic for debate over the past few years: (https://arxiv.org/abs/1506.02142, http://bayesiandeeplearning.org/2016/papers/BDL_4.pdf, https://arxiv.org/abs/1711.02989, ...).
> > > We can clearly show that Dropout does *not* work at estimating higher-order predictive distributions and argue that *this* explains why the method does not translate well to Deep RL (https://arxiv.org/abs/1806.03335).
> > > On the other hand, we show that other Bayesian approaches such as sgmcmc, or ensembles+prior functions *do* work well...
> > >
> > > In terms of rethinking the provocative title... perhaps we could move to add **When** Does Bayesian Deep Learning Work?
> > > We do show very clearly that some methods people have designed to "work" in marginals, do not "work" in the joint... and I think that is a significant and novel contribution.
> > >
> > >
> > > > Discussion of Epistemic/Aleatoric uncertainty and relationship to joint/marginals
> > >
> > > I think that our work highlights a correspondence between Epistemic/Aleatoric uncertainty and joint/marginal predictions.
> > > Our example in Section 2.3 is meant to highlight that what is typically considered through the lens of epistemic/aleatoric uncertainty, can alternatively be understood through the marginal/joint predictive distributions.
> > >
> > > We do not claim to establish a 1:1 equivalence in all settings, however our results (and thought experiment) provide an important counterweight to the influential recent work in the field: https://arxiv.org/abs/1703.04977.
> > > Where that paper suggests we should organise research around epistemic/aleatoric decomposition, we show that a more clear and objective lens can be provided by marginal/joint decompositions.
> > > Even when it comes to flipping a coin... **there is no objective way to determine which part is knowledge, and which is chance**.
> > > The philosophical debate we help to sidestep is whether this should actually matter:
> > > Rather than saying "dropout doesn't model epistemic uncertainty correctly"... we can clearly and objectively say "dropout does not produce good joint predictive distributions.".
> > > Predictive distributions are objective since they depend on the real data, whereas epistemic uncertainty assumes the reference model is "correct".
> > >
> > > Now, I think you raise some good points that there are sentences in the paper that maybe we would be better off to avoid overreaching our concrete.
> > > Clearly, from this review, there are elements of our writing/exposition that are not yet tight.
> > > For example, we can remove/rewrite "We also help to resolve a point of philosophical debate within the deep learning community: the importance of epistemic versus aleatoric uncertainty"...
> > > The key is to emphasize that in terms of *predictions* the choice of epistemic/aleatoric uncertainty is totally irrelevant: difference models can have arbitrary differences in that decomposition and still make the same predictions (and therefore the same decisions downstream).
> > > However, the decomposition into marginal/joint provides a unified, objective and clear way to assess prediction quality.
> > > I think that this work will be of great value to the community overall, even if there are unfinished details on the edges where reasonable people may disagree.

---

> > > > ### Comment · Reviewer_E6jF · 2021-11-18
> > > > **Predictive vs. aleatoric/epistemic**
> > > >
> > > > > Thank you for your continued comments and engagement.
> > > >
> > > > We should use the strength of openreview to allow for actual discussions, hopefully converging to some joint understanding in the end.
> > > >
> > > > I completely agree with your comments on the posterior predictive uncertainty as being the important part of comparing/benchmarking methods, allowing us to ignore whether the underlying model follows A Bayesian gospel _correctly_ or not.
> > > > Your important contribution, in my opinion, is then to demonstrate that even when focusing on the posterior predictive, one needs to be careful to consider the joint instead of the marginal as we lose important performance information otherwise (e.g. the demonstrated dropout problems).
> > > >
> > > > What I still do not see, however, is this continued focus on the epistemic/aleatoric question. The posterior predictive is relevant in the practical application and when comparing models. Aleatoric/epistemic distinctions are only relevant within a fixed model. Conditioned on a specific model, part of my predictive uncertainty can be reduced when I learn more about my environment, and part of it cannot be reduced irrespective of the amount of data I observe. These are indeed highly subjective and have no objective truth to be compared between different models. (Not sure, what you mean with _"epistemic uncertainty assumes the reference model is 'correct'"_. Could you maybe rephrase that?.) E.g. staying with the Kendall & Gal paper you mentioned, one could have a homoscedastic noise model or a heteroscedastic one, as they propose (this focus on a heteroscedastic noise model seems to be also the more  important part of that paper rather than the aleatoric framing, a homoscedastic noise model would also have aleatoric uncertainty, just be boring). Both give potentially different predictive uncertainties that can be compared. Each of the two allows for a separate internal decomposition that one can evaluate to learn more about where the specific model struggles and why, which can then be used, e.g. for active learning, etc.
> > > >
> > > > To summarize. You state that
> > > > > The key is to emphasize that in terms of predictions the choice of epistemic/aleatoric uncertainty is totally irrelevant: difference models can have arbitrary differences in that decomposition and still make the same predictions (and therefore, the same decisions downstream). However, the decomposition into marginal/joint provides a unified, objective and clear way to assess prediction quality.
> > > >
> > > > I completely agree, which is why I am trying to figure out why you try to frame it as a discussion of predictive vs. aleatoric/epistemic when they serve different tasks.
> > > >
> > > > _Procedural Footnote: I am slowly converging to the opinion that we seem to agree to 80+%, and probably also on the remaining parts, we just view them from different directions. I will give your answers a fresh look tomorrow to see whether we do not actually mean the same, just use different vocabulary. So feel free to take this answer as preliminary thoughts that explore the interactivity of openreview._

---

> > > > > ### Author Response · Authors · 2021-11-22
> > > > > **Ongoing conversation**
> > > > >
> > > > > Thanks again - yes this is definitely the benefit of OpenReview!
> > > > >
> > > > > FWIW I do like the summary that you've given here at a high level.
> > > > > Now, probably I would stop short of saying *totally* irrelevant... just because that seems like there could be aspects to active learning / information digestion that might move beyond these predictions right now.
> > > > > However, I view that as more of a pedantic issue and really the core of what we meant to talk about.
> > > > >
> > > > > My suspicion is that there could be a few words/sentences in our paper where we have inadvertently used language that confused/obscured our main message... and I think this idea of viewing things from different directions is a good way to explain some of this.
> > > > >
> > > > > As such, maybe there are specific things that we can target to rephrase/remove from our paper in a refresh version... since I think there are likely specific pieces that contributed to our work not landing as intended.

---

> > > > > > ### Author Response · Authors · 2021-12-01
> > > > > > **Hope for advocacy**
> > > > > >
> > > > > > Hello again!
> > > > > >
> > > > > > As the review process has gone on it seems like all the other reviewers have remained resolute to keep their scores unchanged.
> > > > > > We hold out some hope that you might act as some kind of advocate for this paper. :D
> > > > > >
> > > > > > In some of our other responses we have added evaluation of intermediate tau, together with expected calibration error.
> > > > > > Overall, we still think the main contributions of this paper would be of great value to the community.
> > > > > >
> > > > > > Best wishes

---

### Official Review · Reviewer_QqqV · 2021-11-01

**Correctness:** 4
**Technical Novelty And Significance:** 2
**Empirical Novelty And Significance:** 2
**Recommendation:** 5
**Confidence:** 4

**Main Review:**

I appreciate that benchmark emphasizes joint predictions as an important component in evalauting uncertainty estimates. I agree that these joint predictions are very important for certain applications of uncertainty estimation, especially in sequential d
ecision making settings. However, I am not convinced this benchmark will be a significant and useful contribution to the community in its current state, for the reasons below.

**Existing benchmarks**: Wang et al 2021 [1] already emphasize the differences between joint and marginal uncertainties and propose various ways of evaluating joint predictions, including a downstream transductive active learning task. Interestingly, Wang et al find that directly evaluating joint likelihoods gave little more information than simply marginals, while this paper concludes that different approximate inference methods can produce very different joint likelihoods with similar marginal performance. I suspect the difference here is in the sequence length $\tau$ used to evaluate joint predictions; Wang et al appear to only evaluate over 5 data points, while this paper uses 100 to show major performance differences. It would be good to include additional results at different $\tau$'s between 1 and 100 to see how the difference between joint and marginal predictions grows.

Another benchmark proposed by Wilson et al 2021 [2] also focus on evaluating the faithfulness of Bayesian posterior approximations, though they only evaluate marginal uncertainties. Nontheless, I find their benchmark has several advantanges: they study much larger scale networks (albeit this implies much higher computational cost) and also consider real world input distributions instead of only synthetic ones.

**Regarding likelihoods as the evaluation metric**: While (joint) likelihoods are certainly a very natural choice for comparing predictions, what ultimately matters is how useful the predictions are in making downstream decisions. In addition to just measuring likelihoods, I think the benchmark would be a much stronger contribution if it also emphasized comparisons on downstream tasks, which can be synthetically generated similarly to the existing settings.
Examples of possible downstream tasks that really test the quality of joint predictions could include active learning, contextual bandit, and Bayesian optimization tasks, plugging different posterior approximations into standard representative algorithms for each task (though Wang et al 2021 already do propose transductive active learning as a benchmark task for joint uncertainties).
Another set of potential tasks focusing more on the quality of marginal predictions could include tasks like selective classification, possibly in conjuction with different levels of synethetic covariate shift. Being able to accurately gauge the (relative) trustworthiness of predictions (especially with distribution shift) is certainly an important task for uncertainty estimation, even if it does not rely on the quality of joint predictions.
Within the scope of synethetically generated tasks, I think the inclusion of additional evaluations on a variety downstream tasks into a single consolidated benchmark would be a much stronger contribution, and emphasize how different applications of uncertainty estimation can have different needs and be more amenable to different algorithms.

**Regarding trying to approximate true Bayesian posteriors:** Another hesitation I have is that the focus is that the target is given by quality is being measured against a ground truth consisting of random MLPs, whlie it is hard to say whether these pri
ors actually perform well when faced with real world data and tasks. For example, in Izmailov et al [3], they find that (in larger scale settings), exhaustively running HMC, with presumably better approximations of the true posterior, actually performs worse than approximate methods like ensembling when faced with distribution shift, suggesting simply trying to faithfully approximate exact inference is not necessarily the right (or at least only) goal we consider.

**Scale and synthetic nature of the benchmark**: I also have reservations about the scale of the proposed benchmark, as it is focused on a small data regime with very simple models (MLPs). While the authors state that the synthetic results are predictive
 of real world data, the real world datasets evaluated are also quite toy in nature, and extending insights to larger scale models and datasets like in [2] is quite important.

**Regarding likelihood evaluation in Algo 4**: I believe it would be helpful provide more explanation of Algorithm 4 in appendix A.1. Currently, there is little motivation for the different steps in the algorithm, and I would appreciate greatly if the authors could summarize the relevant concepts from high-dimensional geometry as applied here. Given the paper is proposing a new set of evaluations, it would be especially important to make sure that users have confidence that the metrics used are meaningful. If no formal analysis of the accuracy of the approximation is feasible, there should at least experimental comparisons with the direct Monte Carlo estimate (run with exhaustively many samples to estimate the ground truth) to compare the sample complexities of the proposed estimate at different sequence lengths, as well as documentation of how hyperparameters of the estimator were or should be determined.

Citations:

[1] Wang, Chaoqi, Shengyang Sun, and Roger Grosse. "Beyond Marginal Uncertainty: How Accurately can Bayesian Regression Models Estimate Posterior Predictive Correlations?." International Conference on Artificial Intelligence and Statistics. PMLR, 2021.

[2] Andrew Gordon Wilson, Pavel Izmailov, Matthew D Hoffman, Yarin Gal, Yingzhen Li, Melanie F Pradier, Sharad Vikram, Andrew Foong, Sanae Lotfi, Sebastian Farquhar. "Evaluating Approximate Inference in Bayesian Deep Learning." https://izmailovpavel.github.io/neurips_bdl_competition/.

[3] Izmailov, Pavel, et al. "What Are Bayesian Neural Network Posteriors Really Like?." arXiv preprint arXiv:2104.14421 (2021).
~

**Summary Of The Paper:**

The paper proposes a new benchmark for approximate inference methods, with emphasis on comparing the quality of joint predictions instead of marginals. The proposed testbed is comprised of synethetic test functions generated by simple MLPs, and authors show results on the synthetic benchmark are well correlated with results on small scale real world datasets as well.

**Summary Of The Review:**

I believe the scope of the benchmark is quite limited, and the paper does not show any strong benefits provided by this benchmark over other already existing benchmarks.

---

> ### Author Response · Authors · 2021-11-16
> **Author response**
>
> Thank you very much for your review.
> We are pleased that you agree with the importance of joint predictions, particularly for sequential decision tasks.
> We do believe that this benchmark, which clearly highlights that joint predictions can show something that is hidden by looking at "marginal only", can be of value to the field.
>
> We hope that our main rebuttal address many of the concerns that you raise.
> In particular, we believe that points (1), (3), (4) and (5) are particularly relevant to your review.
>
> Regarding Algorithm 4, we now have some extra space in the paper and should be able to expand a bit on the intuition/high level framing of our algorithm for computing joint likelihood.
> However, it is likely that a fully rigorous treatment of this material would require at least a full conference paper to investigate.
> We are already working on this point, but will not be able to include all of thie technical development in this conference paper.
>
> Overall, we believe that the your summary misses perhaps the key point of this paper.
> We present a new benchmark that investigate the joint predictive distributions beyond the marginal *and* we show that this perspective can have huge difference in ranking state-of-the-art approaches to Bayesian deep learning.
> This occurs even when the underlying model is a simple/canonical MLP and even when all the methods perform similarly in marginals (tau=1).
> We believe that this is a really stimulating and interesting development in a field that is often dominated by small tweaks and training details (see e.g. https://arxiv.org/abs/2110.00476).
>
> We hope that, in the course of this rebuttal period, we can convert your review to become an advocate of this paper at ICLR.
>
> Best wishes.

---

> > ### Comment · Reviewer_QqqV · 2021-11-23
> > **Overall opinion still unchanged**
> >
> > The rebuttal does not currently change my overall opinion on the paper. For the most part, the rebuttal acknowledges drawbacks/limitations of this work (small scale/toy setup, not directly including downstream tasks), but I still feel these drawbacks do make the contribution too limited. Some additional specific points are below.
> >
> > **Regarding discussion of Wang et al 2021**: I again emphasize that Wang et al 2021 already emphasized the importance of joint predictions over marginal predictions. While they find that joint likelihoods are dominated by the margina likelihoods and instead propose a 'cross-normalized log likelihood' instead, I suspect that this is an artifact of the small sequence length $\tau=5$ they were evaluating with, and I think it would especially important in this paper to show results with intermediate $\tau$'s and highlight the reason for this discrepancy (if it does turn out to be due to the small $\tau$). Regardless, this paper does not contradict their main point considering joint predictions is important for downstream tasks, which is shared with this paper. I also do believe that Wang et al should be discussed in related work beyond just saying that you refute their claim that joint likelihoods aren't distinguishable from marginals, as the overall message is very similar.
> >
> > **Regarding algorithm 4**: I understand a formal technical analysis would be out of scope, and look forward to seeing more intuition on the algorithm as well as discussion on related work in high-dimensional probability the algorithm is built upon. I do again emphasize that there should also be empirical validation that the proposed algorithm gives consistent results with extensive Monte Carlo simulation.
> >
> > **Minor point regarding title**: I agree with reviewer E6jF that the title is a bit too attention-grabbing, and a title focused on joint vs marginal predictions would be more appropriate. To me, the question of whether "Bayesian deep learning works" without any additional qualifications really should center on evaluation in downstream applications.

---

> > > ### Author Response · Authors · 2021-11-27
> > > **Author Response**
> > >
> > > This reviewer highlights *limitations* of the work.  We think it would be a useful exercise to focus on *contributions* of the work.  This reviewer emphasizes Wang et al 2021 as similar work.  We discuss here our contributions relative to that paper in order to put things in perspective.
> > >
> > > **DIFFERENTIATING METHODS BASED ON JOINT PREDICTIONS**
> > >
> > > As the reviewer says, at a high level, both Wang et al and our paper try to make the point that considering joint predictions is important.  However, our paper develops a direct measure of accuracy (high-order KL-divergence, or equivalently, log-likelihood) that decisively differentiates methods that do or do not learn accurate joint predictions.  On the other hand, Wang et al makes the contradictory and incorrect statement that “joint log-likelihood scores [are] determined almost entirely by the marginal log-likelihood scores...in practice, they provide little new information beyond marginal likelihoods.”  They may have been misled by focussing on a limited range of experiments for which there was ample training data and therefore epistemic uncertainty played a limited role.  A possible additional factor brought up in the review was their focus on evaluating only up to fifth-order joint predictive distributions; we comment further on this issue below.
> > >
> > > Wang et al go on to produce an alternative metric – the cross-normalized log-likelihood, which depends on a heuristically chosen reference model.  The latter makes the method less systematic than one would hope and makes the results questionable to us.  Further, their metric depends only on second-order statistics and therefore can not capture arbitrarily complex interdependencies that are reflected in KL-divergence.
> > >
> > > Wang et al also assess methods based on downstream task performance.  They focus on active learning, which represents a problem of exploration.  Indeed, exploration algorithms can benefit from accurate joint predictive distributions.  However, our inclination to focus on a pure accuracy metric stems from our own empirical experience.  In particular, we have found that by tuning hyperparameters, one can produce highly inaccurate predictive distributions that work well for a specific exploration algorithm while the same method and hyperparameters fail miserably when coupled with other exploration algorithms.  Our decision to focus on KL-divergence is motivated by this and results in decision theory, which imply that faring well in terms of KL-divergence bounds performance in *any* downstream task.
> > >
> > > **PREDICTIVE DISTRIBUTION ORDER**
> > >
> > > In order to address questions raised in the review concerning how performance comparisons depend on the order tau of the joint predictive distribution, we provide results over the range of 1 through 100 in a plot, which can be accessed via the following anonymous link:
> > >
> > > https://i.postimg.cc/FzBmnTrQ/normalized-kl-tau.png
> > >
> > > This figure plots the normalized KL estimates of ensemble, ensemble+, and mlp for tau=1, 3, 5, 10, 30, 50, 100.  Note that the performance gap between ensemble+ and ensemble (or mlp) increases with tau for tau<= 30 but seems to flatten after that.  Among other things, these results demonstrate that the joint log-likelihood clearly distinguishes between methods even when tau=5.  As such, the claim made in Wang et al that joint log-likelihoods do not provide significant information beyond marginal log-likelihoods can not simply be due to their focus on the range tau<=5.  Rather, there must be other limitations to experiments covered in that paper.
> > >
> > > **REGRESSION VS CLASSIFICATION**
> > >
> > > The work of Wang et al is limited to regression problems.  Indeed, it is not clear how to extend cross-normalized log-likelihood to classification problems.  Our use of KL-divergence, on the other hand, applies to both regression and classification problems in a unified manner.  And our computational results all involve classification problems.  Developing a tractable methodology that applies to classification problems is notably more challenging and also more relevant to common use cases of deep learning.

---

> > > > ### Author Response · Authors · 2021-11-27
> > > > **Response About Title**
> > > >
> > > > While we understand the reviewer's choice on how to interpret the title, we should mention that we intended a different interpretation that we thought could be clear from the abstract and content.  The question we posed was meant to be “how do various Bayesian deep learning methods proposed in the literature perform relative to one another and how should we assess how well each works?”  That would obviously be too long of a subtitle, so we settled on “Does Bayesian Deep Learning Work?”  We did not intend that to be a question about *all* of Bayesian deep learning but rather on how to determine how well specific Bayesian deep learning methods work.  Perhaps we should change the title to “Evaluating Predictive Distributions: Comparing Bayesian Deep Learning Algorithms.”

---

### Official Review · Reviewer_Etjg · 2021-11-02

**Correctness:** 4
**Technical Novelty And Significance:** 3
**Empirical Novelty And Significance:** 4
**Recommendation:** 6
**Confidence:** 4

**Main Review:**

Pros:

Bayesian deep learning is an important research area but the evaluations of the progress made in the field has not been consistent so far which makes it difficult to evaluate different techniques. Furthermore some of the common practice on evaluations on out of distribution data provides insight but lacks providing a quantitative benchmark. The paper addresses an important problem in Bayesian deep learning and provides a framework and tools to evaluate different techniques. The framework is based on simulations which helps provides control over variables and enables gathering insights about various techniques. Furthermore, the paper extend the evaluations to sequential decisions which makes the contributions unique.

Overall I found the contributions of the paper very important for the field. I also think the paper is very written and organized.

Cons:

Even though in the abstract the authors mention that the proposed method provides insights into aleatory and epistemic uncertainty, the manuscript does not elaborate on this point. Even without this point, I think the paper is interesting and removing this point will not hurt its reach.

Bayesian deep learning researchers are in general interested in out of distribution generalization. I found the discussion on OOD a bit limited. I believe the proposed framework can also enable evaluations with OOD and I would recommend authors adding a discussion on this.

Minor comments:
- Abstract: Please explicitly define what you mean by "joint predictions".
- Figure 6. Would be interesting to color code the agents in the Figure.

**Summary Of The Paper:**

The paper proposes a simulation based framework to evaluate different techniques proposed for uncertainty estimation of predictive models. The approach relies on simulated data to control effects such as environments where data is collected, data and model uncertainty. This control enables generating interesting insights about various techniques.

 The paper also go beyond evaluating marginal posterior predictive distributions and extend their benchmarking work to joint distributions capturing sequential decisions that can be made with such models. Some of the highlights from the results: 1) Their results show that Bayesian deep learning is impactful for capturing the joint predictive distributions. 2) Priors used in ensemble+ help with diversity and therefore enable better predictive distributions. 3) Bootstrapping help with robustness of predictions if the models hyperparameters are not tuned.

**Summary Of The Review:**

Overall, I vote for accepting. I like the idea of using simulated settings with explicit control on variables to perform benchmark analysis and this paper enables this through a novel framework as well as open source tools.

---

> ### Author Response · Authors · 2021-11-16
> **Author response**
>
> Thank you very much for your review.
> We are very happy that you "found the contributions of the paper very important for the field" and the paper very well written and organised.
> Given that our reviews seem to be essentially borderline, we might end up relying on you as our potential "champion" of this paper.
>
> We hope that our main rebuttal addresses some of the key weaknesses you have found in the paper.
> In particular, we hope that points (2) and (4) help to assuage some of your main concerns with this work.
> In particular, we think we can make good on our claims re: epistemic/aleatoric uncertainty with just some minor additions to writing.
> Now that we have fixed our formatting issues we have some extra page-space to work with too!
>
> Regarding our discussion on OOD evaluation, we agree that this is limited or rather non-existent in this paper.
> We view this as an important area for future work, but think that there are enough potentially stimulating ideas in this paper to merit acceptance even without that included in this version.
> We believe this is somewhat related to point (4) in our evaluation, where OOD evaluation might be thought of as an important downstream task.
>
> Minor comments:
> - Will try and be more explicit in the abstract.
> - This is easy to do thanks.
>
> Overall, we hope to incorporate your feedback and make this an even more compelling paper for ICLR.
>
> Many thanks.

---

> > ### Comment · Reviewer_Etjg · 2021-11-29
> > **Concerns remain on epistemic/aleatory uncertainty claims**
> >
> > I thank the authors for their comments! I believe there are merits to the benchmarks proposed in the paper through evaluations of the joint posterior predictive distributions. However, I am still not convinced on epistemic/aleatory claims for the paper. I have read your response and especially the discussion with Reviewer E6jF, who shares similar concerns with me on this point. Therefore, I will be keeping my score as weak-accept.

---

> > > ### Author Response · Authors · 2021-12-01
> > > **Thank you**
> > >
> > > Thanks again for your attention on this matter.
> > >
> > > My belief is that Reviewer E6jF may actually be considering to revise their score up, since we realise that some of the issues come down to "viewing the same issue but with slightly different wording".
> > > We hope that, if you they do decide to advocate for this paper maybe that you will also move with them.
> > >
> > > Many thanks

---

### Official Review · Reviewer_4oj5 · 2021-11-05

**Correctness:** 3
**Technical Novelty And Significance:** 3
**Empirical Novelty And Significance:** 3
**Recommendation:** 5
**Confidence:** 3

**Main Review:**

Strengths:

- This paper is generally well-written and clear. Evaluating predictive distributions is an important and relevant topic to the general UQ community.
- As far as I can tell, the idea of evaluating predictive distributions via joint distributions over the numerous predictions is quite novel, as well as the proposed algorithm.
- The experiments section does a good job in demonstrating the discrepancies that can arise between marginal and joint predictions during evaluation, and these findings are interesting.

Weaknesses:

- As this work propose an evaluation metric, I believe it's important to have a discussion on the suite of metrics that are currently used in UQ and how the proposed metric provides an advantage over existing ones. This paper only touches upon KL-divergence (equivalently, cross-entropy, or likelihood), and doesn't mention other widely used metrics, such as calibration, or other proper scoring rules.
- In fact, would calibration (+multiples notions exist, e.g. ECE, classwise-ECE, adaptive ECE, ...)  of the marginal distributions also fail to differentiate between the different methods in Figure 2?
- What is the significance of using $\tau=100$ for the joint predictions? Would there be discrepancies in the ranking of methods based on other $\tau$'s? In practice, if this metric was used for model selection, which $\tau$ should the practitioner base the metric on to choose a model?
- On the same note, I believe there needs to be more demonstration/elaboration that a model with better joint predictions is "necessarily better" than another model that has worse joint predictions, but possibly as good or better marginal predictions. In the second paragraph of page 2, the authors hint that some Bayesian models are not performant in sequential decision tasks because they have poor joint predictions. The proposed metric would be more convincing if this point was followed up with, or if other downstream tasks were demonstrated in the experiments.

Questions on content:

- In Section 4.1, what does the environment variable $\varepsilon$ represent, concretely? Is it the choice of SNR and the initialization distribution over the data generating NN? Let me know if I am missing something here.
- In the last sentence of the second paragraph of Section 4.1, I don't understand what is meant by "agent is provided with the data generating process"? How is this done, concretely?
- In Step 1 of Algorithm 4, what exactly is the distribution, $\mathbb{P}(\hat{\varepsilon} | f_{\theta_{T}})$? How is this distribution produced?

Other points:

- The font and spaces between lines for this submission are quite different from the default style. However, as far as I can tell, it seems to reduce space for content, so I did not flag it.
- Table 1 protrudes beyond the side margins — I'm not sure if this isn't a formatting violation.
- In Figure 6(b), which $\tau$ setting was chosen each of the blue points?

**Summary Of The Paper:**

This work advocates evaluating predictive distributions via joint predictions (rather than the standard practice of evaluating marginal predictions) and introduces The Neural Testbed, an open-source software which includes the testing suite, along with implementations of a handful of methods in uncertainty quantification (UQ). The core evaluation metric used is KL divergence (cross-entropy loss) between the predictive distribution and the true likelihood of the data-generating process, and this work proposes an algorithm to compute this metric with joint distributions. The empirical evaluations compare numerous standard UQ methods with the Testbed, with both synthetic and real datasets.

**Summary Of The Review:**

As noted in the section above, while I do think this is interesting work with interesting findings, I am not fully convinced of the merits of the metrics proposed. Further comparisons and discussion of other existing metrics, along with demonstrations of the benefits of better joint predictions in a downstream task would be helpful in making a stronger case for this work.

---

> ### Author Response · Authors · 2021-11-16
> **Author response**
>
> Thank you very much for your review.
> We are glad that you found the paper well-written and clear, on an important topic and even interesting! :D
>
> We have now uploaded an author response, which outlines some of the key themes raised across reviewers.
> In particular, we hope that points (1), (4) and (5) can help assuage the main concerns you have raised in your review.
>
> To address your content questions:
> - The variable $\mathcal{E}$ represents the specific sampled instance of the neural network generative model.
> - Concretely, the agent is given knowledge of the weight initialization scheme and architecture. We allow the agent to match this distribution, but not the random seeds.
> - This sampling scheme depends on the network architecture but for example in an ensemble, the $f_{\theta_T}$ represents the entire ensemble and sampling $\hat{\mathcal{E}}$ means sampling an element from the ensemble. For something like BBB it means sampling the weights of the Bayesian neural network.
>
> Other points:
> - Thank you for your formatting help, we have rectified the font and that has given us back a lot of space. The table we are working on.
> - Section 5.1 (and figure 6b) is all on marginal predictions tau=1... we will clarify this.
>
> We believe that, through addressing particularly point (4) in our main rebuttal we can convince you that this work will be interesting to the community, and even increase your score.
>
> Best wishes.

---

> > ### Author Response · Authors · 2021-11-29
> > **Author response**
> >
> > **CALIBRATION METRIC**
> >
> > To address the question raised in the review regarding whether calibration metrics of the marginal distributions also fail to differentiate between the different methods in Figure 2, we have provided results for Normalized ECE (Expected Calibration Error normalized by the ECE of the mlp agent), which can be accessed via the following anonymous link:
> >
> > https://i.postimg.cc/1tSzxndR/normalized-expected-calibration-error.png
> >
> > This result clearly demonstrates that ECE also fails to differentiate different methods.
> >
> >
> > **SIGNIFICANCE OF TAU=100**
> >
> >
> > To address the other question raised by the reviewer about the significance of using tau=100 for the joint predictions, we have provided results over the range of 1 through 100 for tau in a plot, which can be accessed via the following anonymous link:
> >
> > https://i.postimg.cc/FzBmnTrQ/normalized-kl-tau.png
> >
> > This figure plots the normalized KL estimates of ensemble, ensemble+, and mlp for tau=1, 3, 5, 10, 30, 50, 100.  Note that the performance gap between ensemble+ and ensemble (or mlp) increases with tau for tau<=30 but seems to flatten after that. Therefore, the ranking of the methods would not change even if we choose a smaller value like 30 for tau.

---

> > > ### Comment · Reviewer_4oj5 · 2021-11-30
> > > **Thanks for the clarifications**
> > >
> > > I would like to thank the authors for their response.
> > >
> > > While the additional experiments to report calibration metrics is very much appreciated, reporting normalized numbers are not all that helpful since the absolute values for calibration error are interpretable and do matter. Further, I do see that the bars all overlap for all methods, which makes me wonder whether the tasks proposed in the Testbed are actually trivial problems for the marginal setting, where differences in the evaluations only arise in the joint setting. I raise this point since there exist prior work that do include many of the baselines evaluated in this work where calibration metrics do differentiate the methods (e.g. [1] below).
> > > Even in my own experience, I have observed very different calibration performance among many of the baseline methods, even for fairly simple benchmark datasets, such as the UCI datasets, so I am quite surprised that all methods perform the same in terms of ECE....
> > >
> > > Many of the other reviewers raised concerns about unnecessary confusion brought about in the content, especially the title, the comic insertion, and the nuances made about Bayesian/non-Bayesian and aleatoric/epistemic uncertainties.
> > > While I overlooked these aspects during my initial review as I was focused on the actual method, I do believe these concerns are valid and reasonable, and would best be addressed with revision in the writing.
> > >
> > > In light of these comments, I will maintain my original score.
> > >
> > > [1] Calibrated Reliable Regression using Maximum Mean Discrepancy (NeurIPS 2020)

---

> > > > ### Author Response · Authors · 2021-12-01
> > > > **Calibration results**
> > > >
> > > > We have now updated results on both varying tau, and evaluating notions of calibration as requested.
> > > >
> > > > Now, it turns out that in the simple generative model proposed in this paper (2-layer MLP) after extensive tuning of each baseline method, each of the agents performs similarly in marginal KL *and* ECE.
> > > > We think it's wrong to penalise our work for not showing large differences in ECE on this problem, particularly since we have opensourced a large quantity of high-quality code to make this fully reproducible.
> > > >
> > > > Regarding *absolute* vs *relative* calibration numbers, these quantities are already averaged over many different seeds, temperatures and data regimes.
> > > > As such, we felt that the the absolute calibration errors were not of much value to include... especially since, as you observe, all the methods perform essentially identically after tuning.
> > > >
> > > > Of course, we do not claim that in *all* datasets, for *all* agents, that there will be no differences in marginal calibration.
> > > > However, we believe that the fact that these results are so simple, clear and clearly *different* from your own experience in e.g. UCI datasets is exactly why this paper is of interest to the community.
> > > > We show clearly, in a maximally-simple and reproducible opensource setting that examining these agents through the lens of *joint* predictions can give drastically different conclusions than looking at marginals alone.
> > > >
> > > > For this reason, we believe that you are overlooking the valuable contributions that this paper can bring to the community.

---

> > > > > ### Author Response · Authors · 2021-12-02
> > > > > **Further plots**
> > > > >
> > > > > To follow up on the requests for un-normalized levels per agent:
> > > > >
> > > > > - accuracy: https://ibb.co/jggDDSZ (all around 80%)
> > > > > - calibration: https://ibb.co/CJMbTHB (all around (0.075)
> > > > >
> > > > > We believe that the average *accuracy* of around 80% shows that the problems are not totally trivial for these agents, since even after tuning they are not able to get all answers correct all the time.

---

### Author Response · Authors · 2021-11-16
**Author response**

## Overall response

First of all, we would like to thank the reviewers for their time and effort during this process. We are happy to note that each review seemed to agree that this is an important area for research, and that our paper is generally well written and presents several novel insights. We believe that this paper, and the accompanying opensource “Neural Testbed” will be stimulate useful discussions within the community and new research directions.

The reviews have already been helpful in raising several questions that follow from this work, some of which we will be able to address in the rebuttal and will be sure to improve our conference paper. Others questions we will have to leave to future work - most notably about whether these insights of joint-vs-marginal “scale up” to real-world downstream tasks and/or larger models. Although we agree these questions are essential for the field as a whole to address, we view the role of this paper to be providing some preliminary (or foundational) inspiration to guide that.

For this reason, we believe that the reviewer scores are somewhat deflated across the board. The reason is that the reviews have focused more on what they correctly identify as “unanswered” questions in this research agenda, and comparatively less on the positive value and insights that this work can generate. We believe that, through this rebuttal process, the reviewers may be encouraged to look more favorably on our work.

## Major themes

We will address the major issues raised across reviews below, with review-specific responses direct to each reviewer to follow.

### (1) Choice of tau=100 and robustness to tau

In our paper we have tried to highlight the potential importance of evaluating predictions beyond marginals, and that this can actually change the answers you get in Bayesian deep learning. We chose tau=100 as a representative of “joint” predictions, to contrast with tau=1 for marginals.

As several reviewers have pointed out, this choice was somewhat arbitrary. We can improve the presentation in the paper by showing how these metrics vary as tau grows from 1,..,100 and beyond. We will add a discussion of this choice to the paper, together with more extensive results/plots in the supplementary appendix. We have found the dependence of the KL divergence on tau to be smooth.

### (2) Insight into the importance of epistemic/aleatoric uncertainty

One of the claims in our abstract is that the paper provides some insight into the role of epistemic/aleatoric uncertainty. However, some of the reviewers did not agree that our paper delivered on this front. We believe that, through improving and expanding our writing we can make the connections/insights more clear.

Our key point is that, when evaluating *predictive uncertainty*, the differences that many models can ascribe to epistemic/aleatoric uncertainty can be *equivalently examined* through their different predictions in joint/marginal predictions. The “biased coin” example of Section 2.3 is designed to highlight exactly this. Here two agents make the same marginal predictions, but with a different attribution to epistemic/aleatoric uncertainty. You cannot tell these agents apart by examining marginals, but the difference becomes clear when examining *joint* predictions.

In fact, we can go further to point out that the distinction between aleatoric/epistemic uncertainty is really dependent on the choice of model, and that there is no objective way to know which is appropriate in the world. We may want to build towards advanced AIs that design *their own model*... as such, it makes more sense to focus on the actual predictions, but to do this effectively you need to look at the *joint* predictions, not just the marginals.

### (3) Toy nature of the problem/models

Several reviewers have pointed out that this paper, and the Neural Testbed are essentially small/toy examples: we agree. However, rather than a significant shortcoming, we believe that it is a strength of this paper to be able to show such clear results in such a simple/foundational model as random MLPs with 2 hidden layers. In a field where many state-of-the-art methods improve test accuracy by fractions of a percentage point, we are able to clearly demonstrate that “state-of-the-art” approaches to Bayesian deep learning in marginals (tau=1), can have completely different performance in joints (tau=100).

Of course, to fully impact the field of deep learning it is essential to investigate/understand how these insights translate from the simple/toy testbed to large-scale models. We believe this is an exciting area for future research, and we would be surprised if these simple MLP models captured all the essential challenges in large scale Bayesian inference. However, we do present some preliminary findings that some of these key insights generalize to real data. We view this as a stimulating starting point for future research, not the “final world” on the topic.

---

> ### Author Response · Authors · 2021-11-16
> **... continued**
>
>
> ### (4) Importance of downstream tasks
>
> Our work is motivated partly by the importance of uncertainty estimation in sequential decision making, but this paper focuses purely on the supervised learning challenge and the importance of joint predictives. Several reviewers have suggested that our case for this importance would be improved by directly showing this relationship to downstream tasks in this paper.
>
> For clarity and conciseness in our limited conference paper, we will not be able to add extensive decision problems to our submission - although we agree this would be interesting follow-on work. To support the importance of “high tau” evaluation we mainly rely on two pieces of work:
>
> - [1] Lu et al https://arxiv.org/abs/2107.09224
> - [2] Osband et al https://arxiv.org/abs/1806.03335
>
> [1] provides an analysis that shows that good joint predictions are sufficient to drive decisions in a bandit problem, but that good marginal predictions are not. We should highlight this more clearly in this submission, and potentially reproduce the analysis in our appendix. [2] provides an empirical evaluation of (mlp, ensemble, dropout, bbb, ensemble+) in conjunction with DQN. The results in this paper show that only ensemble+ is able to perform well in hard exploration tasks, where the other approaches to Bayesian deep learning fail. These results *mirror* our results for tau=100, but cannot easily be explained through looking at marginals alone (where all methods perform similarly).
>
> We hope that by highlighting these connections more explicitly in this paper we can provide more context/evidence for the relevance/insight afforded by evaluation beyond marginals, without significantly changing the experiments we run in this short paper.
>
> ### (5) Comparison to existing metrics/benchmarks
>
> Our paper focuses mainly on KL-divergence (equivalently cross-entropy/NLL), several reviewers have suggested that we should add a discussion of other metrics common to the uncertainty literature (calibration error, brier score, …). We chose KL divergence as a fundamental metric of probability similarities and tried to maintain a simple/clear focus for a short paper. However, adding these metrics will clearly help our reviewers and will be easy for us to accomplish.
>
> We can say from our results internally that the differences in performance joint/marginal is *not* captured by alternative metrics (e.g. ECE, …) when applied to the marginals. We will be sure to add a  clear demonstration of this to our paper. We believe that, in principle, it should be possible to extend the key observations made in our paper to joint versions of these alternative metrics, but leave that to future work.
>
> For comparison to the past work on “cross-normalized log likelihood” (Wang et al), there is little experimental evidence to benchmark against beyond their assertion that joint log likelihoods little additional benefit beyond marginals. In fact, we view our work as a clear refutation of this claim, since we clearly show that methods which perform similarly in marginal predictions can be very different in terms of joints. They actually seem to agree with our assertion that estimating the joint-likelihood *directly* is the more natural approach, and we hope that our work can provide inspiration for more work in this area.

---

### Decision · Program_Chairs · 2022-01-20

**Decision:**

Reject

**Comment:**

The paper describes a new testbed to evaluate Bayesian techniques in the context of joint predictive distribution.  Since this is not the first paper that considers marginal vs joint distribution evaluation, the paper should include a thorough discussion of the differences with prior work.  The paper simply states that it refutes Wang et al.'s previous observation that joint distributions do not distinguish techniques much more than marginals.  However, the paper does not really explain why their observation is correct and Wang's observation should be discarded.  Since this is the core contribution of the paper and it is doubtful, this is problematic.  The discussion of epistemic/aleatoric uncertainty also seems superfluous and therefore distract the reader.